# Sassha: Sharpness-aware Adaptive Second-order Optimization with Stable Hessian Approximation

## Abstract

Approximate second-order optimization methods have gained attention due to their low computational and memory overhead. While these methods have the potential to accelerate neural network training, they often exhibit poorer generalization compared to first-order approaches. To address this limitation, we first analyze existing second-order methods through the lens of the loss landscape, demonstrating that their reduced generalization performance is somewhat attributed to the sharpness of the solutions they converge to. In response, we introduce Sassha, a novel approach designed to enhance generalization by explicitly reducing sharpness. In fact, this sharpness minimization scheme is designed to accommodate lazy and stable Hessian updates, so as to secure efficiency and robustness besides flatness. To validate its effectiveness, we conduct a wide range of deep learning experiments including standard vision and language tasks, where Sassha achieves competitive performance. Notably, Sassha demonstrates strong generalization in noisy data settings and significantly outperforms other methods in these scenarios. Additionally, we verify the robustness of Sassha through various ablation studies.

## 1 Introduction

Recently, second-order methods have been gaining interest due to their potential to accelerate the training process (Yao et al., 2021; Liu et al., 2024; Gupta et al., 2018). Through various techniques for efficient estimation of the second-order derivatives, these approximate second-order methods have achieved faster training with minimal computation and memory overhead compared to their first-order counterparts.

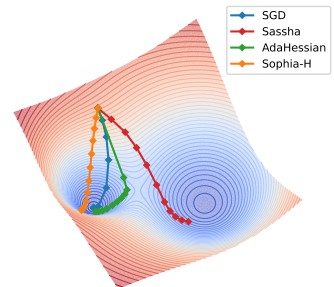

Figure 1: Motivating toy example. Sassha converges to flatter minima compared to other approximate second-order optimizers.

However, contrary to their convergence benefits, recent studies hint at a potentially harmful effect of second-order optimization on generalization. Wadia et al. (2021) argues that second-order optimization impairs generalization by whitening the data. Amari et al. (2021) suggests a more nuanced view; while second-order methods generalize worse under typical conditions, they generalize more robustly in the presence of label noise. Similar observations on deteriorated generalization have also been widely reported for adaptive methods, a closely related class of optimizers that employ preconditioners (Wilson et al., 2017; Zhou et al., 2020; Zou et al., 2022). Despite these observations, there has not been much effort to recover the generalization performance of these optimizers.

Improving generalization remains a central challenge in machine learning, prompting extensive research to better understand underlying factors (Zhang et al., 2017; Neyshabur et al., 2017a). Recent studies have revealed a strong correlation between the flatness of minima and their generalization capabilities (Keskar et al., 2017), spurring the development of optimization techniques aimed at inducing flat minima (Chaudhari et al., 2017; Izmailov et al., 2018; Foret et al., 2021; Orvieto et al., 2022). This line of inquiry has also inspired analyses that attribute the poor generalization of adaptive methods to their tendency to converge to sharp minima (Zhou et al., 2020). Consequently, this raises

an important question: to what type of minima do approximate second-order optimizers converge, and is there potential for improving their generalization performance?

To answer these questions, we first measure the sharpness of different second-order optimizers under various definitions of sharpness, finding that these methods converge to significantly sharper minima compared to stochastic gradient descent (SGD). To improve this, we propose SASSHA— **S**harpness-aware **A**daptive **S**econd-order optimization with **S**table **H**essian **A**pproximation—designed to enhance the generalization of approximate second-order methods efficiently (See Figure 1). Our approach incorporates techniques to stabilize the training dynamics while reducing the computational cost of Hessian approximations. We evaluate SASSHA across diverse vision and natural language tasks, demonstrating that it achieves strong performance relative to existing approximate second-order methods. Moreover, SASSHA shows superior robustness to label noise compared to other practical second-order optimizers and sharpness-aware minimization techniques (Foret et al., 2021). Finally, we conduct a series of ablation studies to provide a comprehensive analysis of our method.

## 2 RELATED WORKS

**Second order optimization for deep learning**    First-order methods such as Stochastic Gradient Descent (SGD) are popular optimization methods for deep learning due to their low per-iteration cost and good generalization (Hardt et al., 2016). However, these methods have two major drawbacks; slow convergence under ill-conditioned landscapes and high sensitivity to hyper-parameter choices such as learning rate (Demeniconi & Chawla, 2020). Adaptive methods (Duchi et al., 2011; Hinton et al., 2012; Kingma & Ba, 2015) propose using empirical Fisher-type preconditioning to alleviate these issues, though recent studies suggest their insufficiency to do so (Kunstner et al., 2019). This has led to recent interest in developing efficient second-order methods tailored for large-scale problems such as Hessian-Free Inexact Newton methods (Martens et al., 2010; Kiros, 2013), stochastic quasi-Newton methods (Byrd et al., 2016; Gower et al., 2016), Gauss-Newton methods (Schraudolph, 2002; Botev et al., 2017), natural gradient methods (Amari et al., 2000), and Kronecker-factored approximations (Martens & Grosse, 2015; Goldfarb et al., 2020). However, a further need for a much scalable second-order optimizer for various large-scale deep learning scenarios has led to much recent focus on using diagonal scaling methods (Bottou et al., 2018; Yao et al., 2021; Liu et al., 2024).

**Sharpness minimization for generalization**    The relationship between the geometry of the loss landscape and the generalization ability of neural networks was first discussed in the work of Hochreiter & Schmidhuber (1994), and the interest in this subject has persisted over time. Expanding on this foundation, subsequent studies have explored the impact of flat regions on generalization both empirically and theoretically (Hochreiter & Schmidhuber, 1997; Keskar et al., 2017; Dziugaite & Roy, 2017; Neyshabur et al., 2017b; Dinh et al., 2017; Jiang et al., 2020). Motivated by this, various approaches have been proposed to achieve flat minima such as regularizing local entropy (Chaudhari et al., 2017), averaging model weights (Izmailov et al., 2018), explicitly regularizing sharpness by solving a min-max problem (Foret et al., 2021), and injecting anti-correlated noise (Orvieto et al., 2022), to name a few. In particular, the sharpness-aware minimization (SAM) (Foret et al., 2021) has attracted significant attention for its strong generalization performance across various domains (Chen et al., 2022; Bahri et al., 2022; Qu et al., 2022) and its robustness to label noise (Baek et al., 2024). Nevertheless, to our knowledge, the sharpness minimization scheme has not been studied to enable second-order methods to find flat minima as of yet.

## 3 PRACTICAL SECOND-ORDER OPTIMIZERS CONVERGE TO SHARP MINIMA

Second-order optimizers have seen rising interest in the deep learning community, and yet, crucial properties of their optimization process remain largely underexplored compared to their first-order counterparts. In particular, SGD and its bias towards the minima of low sharpness have been studied extensively in recent years, which has revealed a strong correlation with its remarkable generalization performance (Keskar et al., 2017; Ghorbani et al., 2019; Wu et al., 2022; Xie et al., 2020). This raises the following question: what minima do second-order optimizers prefer, and how do they correlate with their generalization capability? In this section, we examine through various sharpness metrics employed in recent studies and analyze their correlation with generalization performance.

To measure sharpness, we introduce four metrics frequently used in the literature: maximum eigenvalue of the Hessian, the trace of Hessian, worst-case sharpness, and average sharpness (Hochreiter & Schmidhuber, 1997; Jastrzębski et al., 2018; Xie et al., 2020; Du et al., 2022b; Chen et al.,

Table 1: Sharpness measurements in terms of the maximum eigenvalue $\lambda_{max}(H)$ and the trace $\text{tr}(H)$ of Hessian, worst-case sharpness $\delta L_{\text{worst}}$, and average sharpness $\delta L_{\text{avg}}$ alongside with the generalization error $L_{\text{val}} - L_{\text{train}}$ and the validation accuracy $\text{Acc}_{\text{val}}$ of the solution found by six different optimizers on ResNet-32 trained on CIFAR-100. Approximate second-order optimizers tend to yield minima of high sharpness and worse generalization compared to SGD; SASSHA and M-SASSHA effectively recover this. We provide more results for other workloads from Table 2 in Appendix A where we find the same trends.

| | Sharpness | | | | Generalization | |
|---|---|---|---|---|---|---|
| | $\lambda_{max}(H)$ | $\text{tr}(H)_{\times 10^3}$ | $\delta L_{\text{worst}}$ | $\delta L_{\text{avg} \times 10^{-3}}$ | $L_{\text{val}} - L_{\text{train}}$ | $\text{Acc}_{\text{val}}$ (%) |
| SGD | $265_{\pm 25}$ | $7.29_{\pm 0.30}$ | $0.703_{\pm 0.132}$ | $1.31_{\pm 1.03}$ | $1.027_{\pm 0.013}$ | $69.320_{\pm 0.19}$ |
| Sophia-H | $22797_{\pm 10857}$ | $68.15_{\pm 20.19}$ | $8.13_{\pm 3.082}$ | $19.19_{\pm 6.38}$ | $1.251_{\pm 0.020}$ | $67.760_{\pm 0.37}$ |
| AdaHessian | $11992_{\pm 5779}$ | $46.94_{\pm 17.60}$ | $4.119_{\pm 1.136}$ | $12.50_{\pm 6.08}$ | $0.982_{\pm 0.026}$ | $68.060_{\pm 0.22}$ |
| Shampoo | $436374_{\pm 9017}$ | $6823.34_{\pm 664.65}$ | $73.27_{\pm 12.506}$ | $49307489_{\pm 56979794}$ | $0.508_{\pm 0.07}$ | $64.077_{\pm 0.46}$ |
| M-SASSHA | $382_{\pm 65}$ | $8.75_{\pm 0.31}$ | $2.391_{\pm 0.425}$ | $2.26_{\pm 1.66}$ | $0.628_{\pm 0.010}$ | $70.93_{\pm 0.21}$ |
| SASSHA | $107_{\pm 40}$ | $1.87_{\pm 0.65}$ | $0.238_{\pm 0.088}$ | $0.65_{\pm 0.86}$ | $0.425_{\pm 0.001}$ | $72.143_{\pm 0.16}$ |

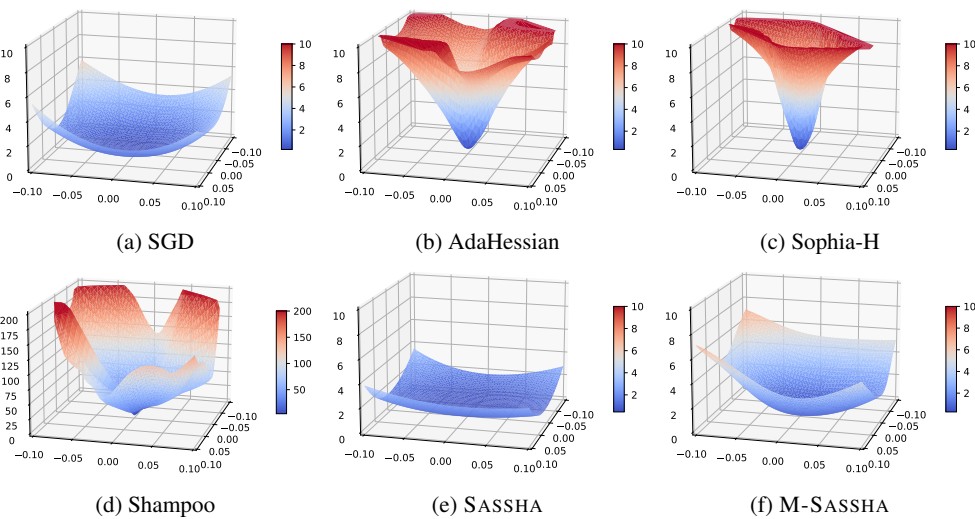

|      |      |      |
|------|------|------|
| (a) SGD | (b) AdaHessian | (c) Sophia-H |
| (d) Shampoo | (e) SASSHA | (f) M-SASSHA |

Figure 2: Loss landscape of minima found by each optimizer on ResNet-32/CIFAR-100 in the direction of dominant eigenvalues. Second-order optimizers can yield minima of extreme sharpness compared to SGD, while SASSHA and M-SASSHA reaches much flatter solution.

2022). The maximum eigenvalue $\lambda_{\max}(H)$ and the trace $\text{tr}(H)$ of the Hessian are often used as standard mathematical measures for the worst-case and the average curvature computed using the power iteration method and the Hutchinson trace estimation, respectively. The other two measures of sharpness are based on the perturbation sensitivity of the loss ($L(x^\star + \epsilon) - L(x^\star)$), where the worst-case sharpness ($\delta L_{\text{worst}}$) is computed with the perturbation maximizing the first-order approximation of the loss function as $\arg\max_{\|\epsilon\| \le \rho} L(x^\star + \epsilon) = \rho \nabla L(x^\star)/\|\nabla L(x^\star)\|$, whereas the average sharpness ($\delta L_{\text{avg}}$) averages the loss difference over Gaussian random perturbation as $\mathbb{E}_{z \sim \mathcal{N}(0,1)}[L(x^\star + \rho z/\|z\|) - L(x^\star)]$. Here we choose $\rho = 0.1$ for the scale of the perturbation.

With these, we measure the sharpness of minima found by three approximate second-order optimizers designed for deep learning; Sophia-H (Liu et al., 2024), AdaHessian (Yao et al., 2021), and Shampoo (Gupta et al., 2018), which we compare to SGD along with our methods, SASSHA and M-SASSHA, on ResNet-32 trained on CIFAR-100. To assess the degree of generalization of these solutions, we also compute the generalization error in terms of loss (*i.e.*, $\Delta L = L_{\text{val}} - L_{\text{train}}$) and the validation accuracy. All experiments are run over three different seeds. We report the results in Table 1.

We observe that existing second-order optimizers can produce solutions with significantly higher sharpness compared to SGD, SASSHA, and M-SASSHA across all definitions of sharpness, which also correlates well with their generalization error. We also visualize the loss landscape of ResNet-32 trained with each optimizer using in the direction of dominant eigenvectors, where we observe sharp minima for second-order optimizers (see Figure 2).

# 4 SASSHA: SHARPNESS-AWARE ADAPTIVE SECOND-ORDER OPTIMIZATION WITH STABLE HESSIAN APPROXIMATION

In the previous section, we observe that the generalization error of approximate second-order methods correlates with the sharpness of the solution. Based on this observation, we propose to incorporate sharpness minimization to improve the generalization of these approximate second-order methods. In the upcoming sections, we present a detailed explanation of various techniques used in SASSHA.

## 4.1 SHARPNESS-AWARE SECOND-ORDER OPTIMIZATION

Many studies have pursued to minimize sharpness during the training process (Chaudhari et al., 2017; Izmailov et al., 2018; Foret et al., 2021; Orvieto et al., 2022). Among them, Foret et al. (2021) propose taking an explicit formulation, which consists of minimizing the objective $f$ within the whole neighborhood of $\rho$-ball through the following min-max problem:

$$\min_{x \in \mathbb{R}^d} \max_{\|\epsilon\|_2 \leq \rho} f(x + \epsilon), \tag{1}$$

where we essentially minimize a slightly perturbed objective within each point in the parameter space to reflect the sharpness of the objective on each position.

Based on this, we construct our sharpness minimization technique for second-order optimization as follows. We first follow a similar procedure as Foret et al. (2021) by solving for $\epsilon$ on the first-order approximation of the objective, which exactly solves the dual norm problem as follows:

$$\epsilon_t^\star = \arg\max_{\|\epsilon\|_2 \leq \rho} f(x_t) + \epsilon^\top \nabla f(x_t) = \arg\max_{\|\epsilon\|_2 \leq \rho} \epsilon^\top \nabla f(x_t) = \rho \frac{\nabla f(x_t)}{\|\nabla f(x_t)\|_2}. \tag{2}$$

We plug this back to yield the following perturbed objective function:

$$\tilde{f}_t(x) \coloneqq f\left(x + \rho \frac{\nabla f(x_t)}{\|\nabla f(x_t)\|_2}\right),$$

which shifts the point of the approximately highest function value within the neighborhood to the current iterate. This essentially penalizes the objective by sharpness; *i.e.* the more drastic the function changes within the neighborhood of the current iterate, the stronger the penalization becomes.

With this sharpness-penalized objective, we proceed to make a second-order Taylor approximation:

$$x_{t+1} = \arg\min_x \tilde{f}_t(x_t) + \nabla \tilde{f}_t(x_t)^\top (x - x_t) + (x - x_t)^\top \tilde{H}_t(x_t)(x - x_t), \tag{3}$$

where $\tilde{H}_t$ denotes the Hessian of $\tilde{f}_t$. Using the first-order optimality condition, we derive the basis update rule for our sharpness-aware second-order optimizer:

$$\begin{aligned}
x_{t+1} &= x_t - \tilde{H}_t(x_t)^{-1} \nabla \tilde{f}_t(x_t) \\
&= x_t - H\left(x_t + \rho \frac{\nabla f(x_t)}{\|\nabla f(x_t)\|_2}\right)^{-1} \nabla f\left(x_t + \rho \frac{\nabla f(x_t)}{\|\nabla f(x_t)\|_2}\right),
\end{aligned} \tag{4}$$

where $H$ denotes the Hessian of the original objective function $f$. Here, instead of directly computing the exact Hessian which is prohibitively expensive, we employ the diagonal approximation of the Hessian estimated via Hutchinson's method (denoted as $\widehat{H}$) with the exponential moving average, which is a standard practice in deep learning since it only requires one additional backpropagation (Yao et al., 2021; Liu et al., 2024).

## 4.2 IMPROVING STABILITY

**Avoiding critical points** The objective of training deep neural networks is known to be highly non-convex with saddle points and local maxima (Dauphin et al., 2014; Choromanska et al., 2015). One problem that arises from naively applying the approach of Section 4.1 to a non-convex objective is that it can ascend in the directions of negative Hessian eigenvalues towards saddle points or local maxima, due to the first-order optimality condition being valid only when all stationary points are minima. While this classical issue of naive second-order optimizers has led to the introduction of

various techniques such as damping (Levenberg, 1944; Marquardt, 1963), positive semi-definite approximations (Amari et al., 2000), clipping (Nocedal & Wright, 1999; Liu et al., 2024), cubic regularization (Nesterov & Polyak, 2006), etc., we follow prior works of Becker et al. (1988); Yao et al. (2021) to apply the absolute function to adjust the negative entries of the diagonal Hessian to be positive, *i.e.*

$$|\widehat{H}| := \sum_{i=1}^{d} |\widehat{H}_{ii}| \mathbf{e}_i \mathbf{e}_i^\top \tag{5}$$

where $\widehat{H}_{ii}$ and $\mathbf{e}_i$ are the $i^{\text{th}}$ diagonal entry of the approximate diagonal Hessian and the $i^{\text{th}}$ standard basis vector, respectively. Here the basic idea is that inverting the sign of the negative eigenvalues will turn saddle points into repellers while achieving the same optimal rescaling as Newton's method (Nocedal & Wright, 1999; Murray, 2010; Dauphin et al., 2014; Wang et al., 2013), which we directly apply to our diagonal Hessian approximation. We empirically validate the effectiveness of this approach via ablation studies in Appendix G.1.

**Alleviating divergence**    Diagonal Hessian estimations are known to sometimes yield overly large steps when they underestimate the curvature (Dauphin et al., 2015). While this generally applies to all approximate second-order optimizers, this instability seems to be more present under sharpness minimization. We believe this is due to smaller top Hessian eigenvalue $\lambda_1$ from sharpness minimization (Agarwala & Dauphin, 2023; Shin et al., 2024) yielding smaller estimated diagonal entries on average:

$$\mathbb{E}\left[\frac{1}{d}\sum_{i=1}^{d}\widehat{H}_{ii}\right] = \frac{1}{d}\sum_{i=1}^{d}\mathbb{E}[\widehat{H}]_{ii} = \frac{\text{tr}(H)}{d} = \frac{1}{d}\sum_{i=1}^{d}\lambda_i \leq \lambda_1,$$

which pushes them closer to zero yielding numerical instabilities during inversion in tandem with curvature underestimations, causing increased training failures. To address this issue, we propose square rooting the Hessian preconditioner, *i.e.*, $|\widehat{H}|^{1/2}$. Its benefit can be understood from two perspectives. First, the square root can alleviate instability from near-zero diagonal Hessian entries by selectively increasing the magnitude of the near-zero diagonal Hessian entries in the denominator (*i.e.*, $h < \sqrt{h}$ if $0 < h < 1$). Also, it can be interpreted as geometrically interpolating the preconditioning matrix toward the identity matrix ($\lim_{\alpha \to 0} H^\alpha = I$), making the optimizer more akin to unpreconditioned first-order methods (Amari et al., 2021). This adjustment weakens the dependency of the optimizer on the preconditioner and helps suppress the influence of any pathological estimations that may occur. We present an empirical analysis of our square-rooted preconditioning in Section 6.1.

### 4.3    IMPROVING EFFICIENCY VIA LAZY HESSIAN UPDATE

While the diagonal Hessian estimator introduced in Section 4.1 significantly reduces the Hessian computations, it still requires at least twice as much backpropagation compared to gradient-based methods. Here we attempt to further alleviate this by lazily computing the Hessian every $k$ steps:

$$D_t = \begin{cases} \beta_2 D_{t-1} + (1 - \beta_2)|\widehat{H}(x_t + \epsilon_t^\star)| & \texttt{if } t \bmod k = 1 \\ D_{t-1} & \texttt{otherwise} \end{cases}, \tag{6}$$

where $D_t$ and $\beta_2$ are the moving average of the Hessian and its hyperparameter. This reduces the overhead from additional Hessian computation by $1/k$ (see Appendix G.3 for detailed cost analysis). Across all evaluations in Section 5 except for language finetuning, we reuse the Hessian estimate for $k = 10$ iterations.

However, extensive Hessian reusing will lead to significant performance degradation since it would no longer accurately reflect the current curvature (Doikov et al., 2023). Interestingly, SASSHA is quite resilient against prolonged reusing, keeping its performance relatively high over longer Hessian reusing compared to other approximate second-order methods. Our investigation reveals that along the trajectory of SASSHA, the Hessian tends to change less frequently over a given number of iterations compared to existing alternatives. We hypothesize that the introduction of sharpness minimization plays an integral role in this phenomenon by biasing the optimization path toward regions with lower curvature change, allowing the prior Hessian to remain relevant over more extended steps. We provide a detailed empirical analysis of the lazy Hessian updating in Section 6.2.

Table 2: Image classification results of various optimization methods in terms of final validation accuracy (mean±std) across different models on the CIFAR-10, CIFAR-100, and ImageNet. SASSHA consistently outperforms others in all settings.

| | CIFAR10 | | CIFAR100 | | ImageNet | |
|---|---|---|---|---|---|---|
| | ResNet-20 | ResNet-32 | ResNet-32 | WRN-28-10 | ResNet-50 | ViT-s-32 |
| SGD | $92.027_{\pm 0.32}$ | $92.693_{\pm 0.06}$ | $69.320_{\pm 0.19}$ | $80.063_{\pm 0.15}$ | $75.581_{\pm 0.05}$ | $62.904_{\pm 0.36}$ |
| AdamW | $92.040_{\pm 0.11}$ | $92.420_{\pm 0.13}$ | $68.783_{\pm 0.22}$ | $79.090_{\pm 0.35}$ | $75.375_{\pm 0.08}$ | $66.459_{\pm 0.15}$ |
| AdaHessian | $92.003_{\pm 0.17}$ | $92.483_{\pm 0.15}$ | $68.060_{\pm 0.22}$ | $76.917_{\pm 0.26}$ | $73.640_{\pm 0.16}$ | $66.417_{\pm 0.23}$ |
| Sophia-H | $91.814_{\pm 0.27}$ | $91.985_{\pm 0.08}$ | $67.760_{\pm 0.37}$ | $79.353_{\pm 0.24}$ | $72.064_{\pm 0.49}$ | $62.436_{\pm 0.36}$ |
| Shampoo | $88.547_{\pm 0.83}$ | $90.227_{\pm 0.24}$ | $64.077_{\pm 0.46}$ | $74.063_{\pm 1.28}$ | * | * |
| SASSHA | $\mathbf{92.983}_{\pm 0.05}$ | $\mathbf{94.093}_{\pm 0.24}$ | $\mathbf{72.143}_{\pm 0.16}$ | $\mathbf{83.543}_{\pm 0.08}$ | $\mathbf{76.429}_{\pm 0.18}$ | $\mathbf{69.195}_{\pm 0.30}$ |
| M-SASSHA | $\underline{92.363}_{\pm 0.23}$ | $\underline{93.177}_{\pm 0.30}$ | $\underline{70.930}_{\pm 0.21}$ | $\underline{81.533}_{\pm 0.27}$ | $\underline{76.004}_{\pm 0.04}$ | $\underline{68.038}_{\pm 0.14}$ |

*Omitted due to computation/memory requirements exceeding available resources.

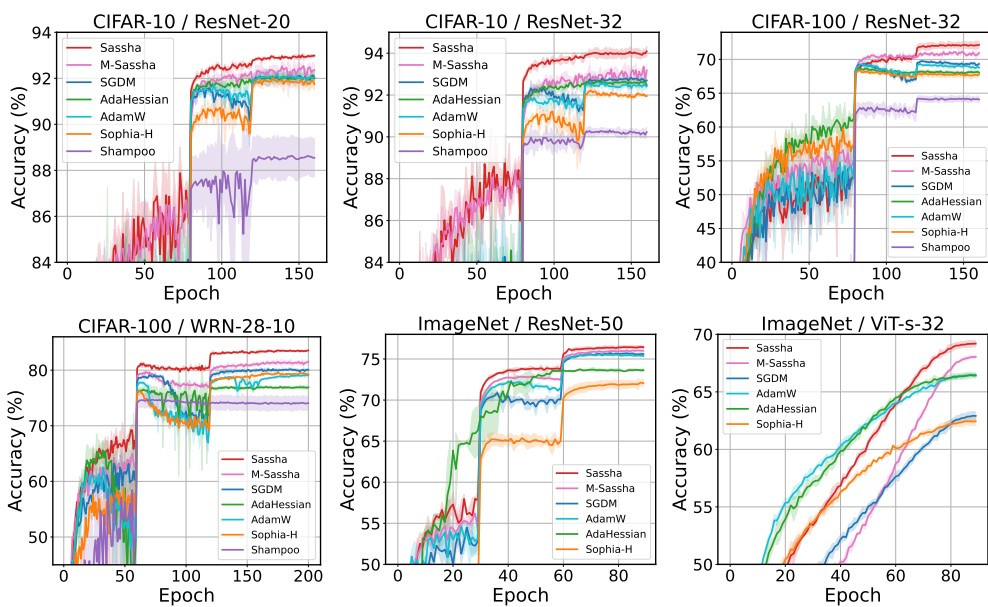

Figure 3: Image classification results for various optimization methods in terms of validation accuracy curve across different models on the CIFAR-10, CIFAR-100, and ImageNet. SASSHA consistently outperforms others in all settings.

### 4.4 FURTHER EXTENSION AND ANALYSES

We further provide simple extensions and supplementary analyses:

1. M-SASSHA, a simple extension to SASSHA that removes additional gradient computation in sharpness minimization and achieves computational costs comparable to first-order methods, is provided in Appendix B.

2. An algorithm table for SASSHA, along with comparisons to other approximate second-order methods, is provided in Appendix C.

3. A preliminary convergence analysis of SASSHA is provided in Appendix D.

## 5 EVALUATIONS

In this section, we compare SASSHA against existing approximate second-order optimizers and demonstrate its superior generalization capabilities. To achieve this, we conduct extensive evaluations across multiple domains, including vision and natural language tasks, as well as under challenging conditions such as label noise for image classification. Specifically, we first compare SASSHA to the following baselines: AdaHessian (Yao et al., 2021), Sophia-H (Liu et al., 2024), AdamW (Loshchilov & Hutter, 2018), Shampoo Gupta et al. (2018), and SGD. We describe experiment settings in detail in Appendix E.

Table 4: Language finetuning results for various optimizers on SqueezeBERT evaluate on the GLUE benchmark. SASSHA achieves better results than AdamW and AdaHessian and compares competitively with Sophia-H.

|  | CoLA M corr. | SST-2 Acc | MRPC Acc / F1 | STS-B S/P corr. | QQP F1 / Acc | MNLI mat/m.mat | QNLI Acc | RTE Acc |
|---|---|---|---|---|---|---|---|---|
| AdamW | 48.63 | 90.25 | 86.27 / 90.21 | 88.36 / 88.40 | 86.63 / 89.96 | 81.20 / 82.20 | 90.13 | 70.04 |
| AdaHessian | 48.36 | 90.60 | 86.52 / 90.43 | 88.72 / 89.01 | 87.41 / 90.65 | 81.15 / 82.10 | 90.02 | 71.12 |
| Sophia-H | **49.12** | **92.32** | 86.03 / 90.19 | 88.87 / 89.13 | 87.70 / 90.84 | 81.68 / **82.54** | **90.26** | 72.56 |
| SASSHA | 47.19 | 91.51 | 87.75 / 91.23 | **89.29 / 89.66** | 87.93 / 91.00 | **81.74** / 82.12 | 89.86 | **73.29** |
| M-SASSHA | 48.84 | 91.28 | **89.23 / 92.22** | 88.29 / 88.39 | 87.91 / 90.97 | 81.64 / 82.09 | 89.84 | 72.92 |

## 5.1 IMAGE CLASSIFICATION

We evaluate SASSHA in comparison to other optimizers on image classification tasks on CIFAR-10, CIFAR-100, and ImageNet datasets (Deng et al., 2009). We train various models including ResNet-20, ResNet-32, ResNet-50 (He et al., 2016), WideResNet-28-10 (WRN-28-10) (Zagoruyko & Komodakis, 2016) and ViT-s-32 (Beyer et al., 2022). We use standard inception-style data augmentations during training without the use of advanced data augmentation techniques (DeVries & Taylor, 2017) or regularization methods (Gastaldi, 2017; Yamada et al., 2019) to focus exclusively on the effect of sharpness minimization. Results are presented in Table 2 and Figure 3. Additionally, we provide the validation loss curves in Appendix F for further insight.

We find that our method achieves the best validation performance across all settings. Particularly, SASSHA achieves a 1% to 4% increase in performance compared to best-performing adaptive or second-order optimizers. Also, M-SASSHA outperforms approximate second-order optimizers by 0.3% to 2% with twice as less computational overheads.

## 5.2 LANGUAGE PRETRAINING

A recent study has demonstrated the potential of approximate second-order methods on pretraining language models (Liu et al., 2024), a core task in modern machine learning for constructing large-scale foundational models. Motivated by this, here we examine how SASSHA compares with existing optimizers on language model pretraining. Specifically, we train GPT1-mini, a scaled-down variant of GPT1 (Radford et al., 2019) with four attention layers instead of the original twelve, with SASSHA and various baseline optimizers on the next word prediction task of Wikitext-2 dataset (Merity et al., 2022) and compare the final test perplexity (refer to Appendix E for detailed experimental settings). The results are presented in Table 3. Our results show that SASSHA achieves the lowest perplexity among all methods, with M-SASSHA following closely, highlighting im-

Table 3: Language pretraining results. SASSHA achieves lower perplexity compared to other second-order methods.

| Optimizer | Perplexity ↓ |
|---|---|
| AdamW | 175.06 |
| AdaHessian | 407.69 |
| Shampoo | 1727.75 |
| Sophia-H | 125.60 |
| SASSHA | **122.40** |
| M-SASSHA | 125.01 |

proved language modeling capabilities. In addition, unlike Sophia-H, the leading baseline, whose performance is largely restricted to its intended language domains (Liu et al., 2024), SASSHA also proves highly effective in image classification (see Table 2).

## 5.3 LANGUAGE FINETUNING

To comprehensively evaluate SASSHA, we extend our experiments to include eight diverse tasks from the GLUE benchmark (Wang et al., 2018). We finetune SqueezeBERT (Iandola et al., 2020) on these tasks and report the final performance on the development set, as presented in Table 4. Our method achieves higher scores compared to other optimizers across nearly all tasks. Notably, M-SASSHA exhibited better performance than AdamW (Loshchilov & Hutter, 2018), the standard optimizer for finetuning language models, on several tasks—even though M-SASSHA has a computational cost similar to AdamW. Additionally, SASSHA records higher scores than Sophia-H (Liu et al., 2024), an optimizer specialized for language models, on many tasks.

Table 5: Robustness to label noise. Here we measure the validation accuracy under various levels of label noise using ResNet-32 trained on CIFAR-100 and CIFAR-10. SASSHA and M-SASSHA shows much robust performance under label noise.

| | CIFAR-100 | | | CIFAR-10 | | |
|---|---|---|---|---|---|---|
| Noise level | 20% | 40% | 60% | 20% | 40% | 60% |
| SGD | $62.18_{\pm0.06}$ | $55.78_{\pm0.55}$ | $45.53_{\pm0.78}$ | $89.91_{\pm0.87}$ | $87.26_{\pm0.4}$ | $82.72_{\pm1.59}$ |
| SAM | $65.53_{\pm0.11}$ | $\underline{61.20}_{\pm0.17}$ | $51.93_{\pm0.47}$ | $\underline{92.27}_{\pm0.14}$ | $\underline{90.11}_{\pm0.25}$ | $\underline{85.79}_{\pm0.30}$ |
| Sophia-H | $62.34_{\pm0.47}$ | $56.54_{\pm0.28}$ | $45.37_{\pm0.27}$ | $89.93_{\pm0.001}$ | $87.30_{\pm0.51}$ | $82.78_{\pm1.43}$ |
| AdaHessian | $63.06_{\pm0.25}$ | $58.37_{\pm0.13}$ | $46.02_{\pm1.96}$ | $90.11_{\pm0.001}$ | $86.88_{\pm0.004}$ | $83.25_{\pm0.004}$ |
| Shampoo | $58.85_{\pm0.66}$ | $53.82_{\pm0.71}$ | $42.91_{\pm0.99}$ | $88.14_{\pm0.29}$ | $85.15_{\pm0.61}$ | $81.16_{\pm0.30}$ |
| SASSHA | $\mathbf{66.78}_{\pm0.47}$ | $\mathbf{61.97}_{\pm0.27}$ | $\mathbf{53.98}_{\pm0.57}$ | $\mathbf{92.49}_{\pm0.11}$ | $\mathbf{90.29}_{\pm0.11}$ | $\mathbf{86.50}_{\pm0.08}$ |
| M-SASSHA | $\underline{66.10}_{\pm0.26}$ | $61.13_{\pm0.28}$ | $\underline{52.45}_{\pm0.34}$ | $91.27_{\pm0.31}$ | $88.85_{\pm0.31}$ | $85.17_{\pm0.24}$ |

## 5.4 ROBUSTNESS TO LABEL NOISE

We evaluate the robustness of our method against label noise (Natarajan et al., 2013), a common issue in real-world scenarios where the training data is incorrectly labeled. To this end, we compare the validation performance of different optimizers on ResNet-32 trained with the CIFAR datasets under various noise levels. The results, summarized in Table 5, show that SASSHA outperforms other optimizers across all noise levels with minimal accuracy degradation; at most outperform by $8\%$ in $60\%$ noise ratio compared to SGD.

Interestingly, our method surpasses SAM (Foret et al., 2021), which is known to be one of the most robust techniques against label noise (Baek et al., 2024). We hypothesize that SASSHA's superior robustness stems from the combined benefits of SAM and second-order methods. Specifically, SAM enhances robustness by applying adversarial perturbations to the weights and giving more importance to clean data during optimization, making the model more resistant to label noise (Foret et al., 2021; Baek et al., 2024). Also, recent research indicates that second-order optimizers are robust to label noise (Amari et al., 2021) due to appropriate preconditioning that reduces the variance caused by label noise in the population risk. We believe these two complementary mechanisms work synergistically within SASSHA to enhance its robustness.

## 6 ABLATIONS

### 6.1 STABILIZING EFFECT OF SQUARE-ROOT

In this study, we examine the stabilizing effect of the square-root function on SASSHA. Precisely, we conduct multiple runs of SASSHA without the square-root (No-Sqrt) over different random seeds for training ResNet-32 on CIFAR-100, which reveals instances where the training loss diverges. To gain further insight into this phenomenon, we measure the update size and the preconditioned step size (i.e., $\|\eta D^{-1}\|_F$, where $D$ is either $|\widehat{H}|^{1/2}$ or $|\widehat{H}|$ and $\|\cdot\|_F$ denotes the Frobenius norm) of each iteration, along with the density of the diagonal precondition entry values $D_{ii}$ at step 100 to 250. Results are presented in Figure 4. We first observe that the training loss diverges precisely when the update size (Figure 4b), particularly the preconditioned step size (Figure 4c), starts to spike around step 200, suggesting that the preconditioner reaches some critical condition at this point. Further investigation into individual preconditioner entries (Figure 4d) reveals that this is likely due to a progressive increase in near-zero diagonal Hessian entries from the sharpness minimization penalizing the Hessian eigenvalues, which could have caused instability when the preconditioner is inverted. With the inclusion of the square-root, we can see that the values within the preconditioner are less situated near zero, effectively suppressing the risk of large updates thereby stabilizing the training process.

### 6.2 ANALYZING LAZY HESSIAN UPDATING

**Effect of Hessian update interval** $k$ ⠀Here we compare how different approximate second-order optimizers, specifically SASSHA, AdaHessian, and Sophia-H, perform under different levels of lazy updating on ResNet-32 trained on CIFAR-100. We vary the update interval $k$ from 1 to 100 and see how each optimizer performs. As shown in Figure 5a, SASSHA consistently outperforms across all

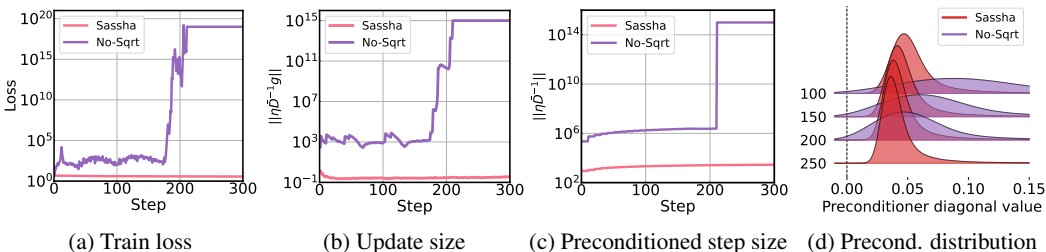

(a) Train loss     (b) Update size     (c) Preconditioned step size     (d) Precond. distribution

Figure 4: Effect of square root on (a) the train loss, (b) the update size, (c) the preconditioned step size, and (d) the distribution of preconditioner values of SASSHA without the square-root (No-Sqrt) and the original SASSHA on ResNet-32/CIFAR-100. The sharpness-minimization alone in No-Sqrt drives diagonal Hessian values towards zero, leading to divergent behaviors. The square-root helps counteract this effect, thereby stabilizing the training of SASSHA.

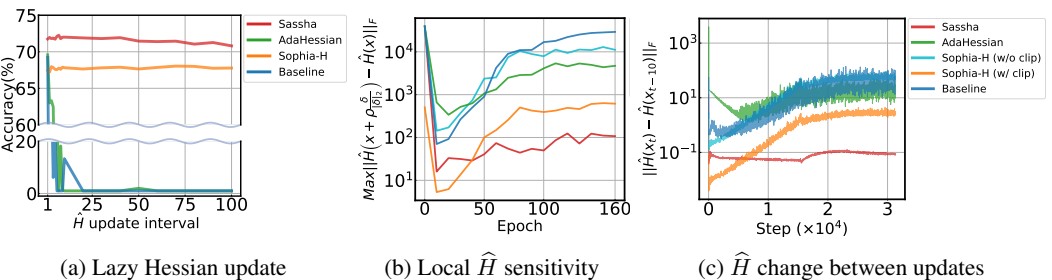

(a) Lazy Hessian update     (b) Local $\widehat{H}$ sensitivity     (c) $\widehat{H}$ change between updates

Figure 5: Effect of sharpness minimization on lazy Hessian training for ResNet-32 trained on CIFAR-100. (a) Ablation of Hessian update interval for SASSHA, SASSHA without sharpness minimization and square-root (Baseline), AdaHessian, and Sophia-H. SASSHA remains effective even with a Hessian update interval of 100. (b) the local Hessian sensitivity from Equation (7) and (c) The norm of the difference of Hessian between every 10 iterations. SASSHA mostly stays within the region where the Hessian is less sensitive.

update intervals, followed by Sophia-H by a margin of almost $5\%$ accuracy difference. AdaHessian is the most vulnerable under lazy Hessian update, showing a rapid decline in performance even with small intervals.

**Sharpness minimization can improve lazy Hessian training**    Why would SASSHA be so robust under a lazy Hessian update? We first reasonably assume that this is due to the presence of sharpness-minimization, as it is the primary component differentiating SASSHA from other approximate second-order optimizers. To verify our hypothesis, we perform the same ablation on the Hessian update interval on the SASSHA without sharpness minimization (Baseline[1]), which we report in Figure 5a. We indeed observe that Baseline doesn't perform as well as SASSHA under lazy Hessian training, showing similar performance to AdaHessian.

Furthermore, we hypothesize that sharpness minimization aids the lazy Hessian update by biasing the optimization path toward the region of low curvature sensitivity. To quantify this, we define local Hessian sensitivity as the maximum change in Hessian induced from normalized random perturbations:

$$\max_{\delta \sim \mathcal{N}(0,1)} \left\| \widehat{H}\left(x + \rho \frac{\delta}{\|\delta\|_2}\right) - \widehat{H}(x) \right\|_F \tag{7}$$

where $x$, $\delta$, $\rho$, and $\| \cdot \|_F$ each denotes a point on the optimization path, the Gaussian random perturbation, the length of the normalized perturbation, and the Frobenius norm, respectively. A smaller Hessian sensitivity would suggest reduced variability in the loss curvature, leading to greater relevance of the current Hessian for subsequent optimization steps.

---

[1]We also remove the square root, as its purpose of stabilization is relevant only in the context of employing sharpness minimization.

To check the sanity of Equation (7), we see if this aligns with a more direct measure of lazy updatability: the Hessian difference between current Hessian and previous Hessian computed $k$ iteration earlier:

$$\|\widehat{H}(x_t) - \widehat{H}(x_{t-k})\|_F. \tag{8}$$

An optimizer with a higher Hessian difference would mean that prior Hessian would easily be outdated, making lazy updating harder to execute. We measure these two metrics with $\rho = 0.1$ and $k = 10$ and report the results in Figure 5b and 5c.

Our results show that SASSHA keeps both of these metrics relatively low throughout most of the training process, while Baseline yields higher Hessian differences. This low Hessian sensitivity significantly improves the reusability of Hessian information compared to AdaHessian, which supports our assumption. However, Sophia, despite its relatively robust lazy Hessian performance, shows a high value of Hessian differences indicative of lower reusability. Interestingly, this difference becomes much smaller when Sophia's per-coordinate Hessian clipping technique is applied to it. This supports the claim by Liu et al. (2024) that clipping, like SASSHA, is also a powerful technique that helps with lazy Hessian updates.

### 6.3 Comparison with SAM

Thus far, our primary focus has centered on validating the effectiveness of SASSHA in the context of approximate second-order optimization. While this remains the principal objective of our study, here we additionally compare SASSHA with SAM to highlight its potential benefits. Specifically, we evaluate two versions of SAM with SGD and AdamW respectively as its base optimizers, and compare them with SASSHA for training ViT-s-32 on ImageNet. The results are provided in Table 6. First, we find that SASSHA performs on par with or better than SAM,

Table 6: SASSHA vs. SAM. SASSHA achieves better performance than SAM even when SAM is allocated more data budgets or longer training time.

|  | Epoch | Time (s) | Accuracy (%) |
|---|---|---|---|
| SAM $_{\text{SGD}}$ | 180 | 220,852 | $65.403_{\pm 0.63}$ |
| SAM $_{\text{AdamW}}$ | 180 | 234,374 | $68.706_{\pm 0.16}$ |
| SASSHA | 90 | 123,948 | $\mathbf{69.195}_{\pm 0.30}$ |

even when SAM is given more data budgets or wall-clock training time. This is potentially due to the benefit of the second-order scheme in SASSHA that accelerates the optimization process. Also, we observe that SAM shows a performance discrepancy between different base optimizers. Notably, one needs to select which base optimizer to use when employing SAM unlike SASSHA. Additional comparisons on other models and datasets are presented in Appendix G.2.

### 6.4 Additional ablations

We provide more ablation results including the effect of the absolute function and a comprehensive cost analysis of SASSHA and M-SASSHA in Appendix G.2.

## 7 Conclusion

In this work, we have addressed the poor generalization issue of approximate second-order methods by proposing SASSHA, which explicitly minimizes sharpness within approximate second-order optimization, achieving competitive performance for various standard deep learning tasks. Nonetheless, there are many remaining possibilities for further improvements which may include, but are not limited to, evaluating on a more extreme scale and other data distributions in different domains, and developing theoretical properties such as convergence rate and implicit bias, all to more rigorously confirm the value of SASSHA. Seeing it as an exciting opportunity, we are planning to investigate further in future work.

## Reproducibility

We made extensive efforts to ensure that our experimental results are reproducible, including providing comprehensive descriptions of all experimental configurations and hyperparameters (Appendix E), details of the hardware used during training and evaluation (Appendix H), as well as the use of, and providing links to, publicly available datasets and source codes for various algorithms used in our experiments. After publication, we also plan to release our source code, which will include algorithm

implementations, hyperparameter settings, dependencies, and hardware configurations necessary for reproduction.

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

# A SHARPNESS MEASURES FOR OTHER SETTINGS

Table 7: The maximum Hessian eigenvalue $\lambda_{max}(H)$, trace of Hessian $\text{tr}(H)$, worst-case sharpness $\delta L_{\text{worst}}$, average sharpness $\delta L_{\text{worst}}$, generalization error in terms of loss $\Delta L$ and accuracy $\Delta\text{acc}$ of the solution found by seven different optimizers on CIFAR-10/100. Second-order optimizers tend to yield minima of high sharpness and worse generalization compared to SGD; SASSHA and M-SASSHA effectively recover this.

| | | Sharpness | | | | Generalization | |
|---|---|---|---|---|---|---|---|
| | | $\lambda_{max}(H)$ | $\text{tr}(H)_{\times 10^3}$ | $\delta L_{\text{worst}}$ | $\delta L_{\text{avg} \times 10^{-3}}$ | $\Delta L(L_{\text{val}} - L_{\text{train}})$ | $\text{Acc}_{\text{val}}$ |
| CIFAR-10 | | | | | | | |
| ResNet20 | SGD | $107_{\pm 4.37}$ | $1.38_{\pm 0.01}$ | $0.840_{\pm 0.304}$ | $0.690_{\pm 0.39}$ | $0.27_{\pm 0.007}$ | $92.027_{\pm 0.32}$ |
| | SAM | $58_{\pm 2.98}$ | $0.73_{\pm 0.04}$ | $0.171_{\pm 0.038}$ | $0.461_{\pm 0.24}$ | $0.119_{\pm 0.002}$ | $92.847_{\pm 0.07}$ |
| | Sophia-H | $3606_{\pm 303}$ | $31.24_{\pm 2.628}$ | $6.120_{\pm 1.634}$ | $18.11_{\pm 1}$ | $0.28_{\pm 0.009}$ | $91.814_{\pm 0.27}$ |
| | AdaHessian | $23048_{\pm 29932}$ | $189.48_{\pm 240.55}$ | $4.538_{\pm 1.634}$ | $198.66_{\pm 266}$ | $0.199_{\pm 0.023}$ | $92.003_{\pm 0.17}$ |
| | Shampoo | $647066_{\pm 419964}$ | $3899.5_{\pm 1825}$ | $166.3_{\pm 48.0}$ | $2177189_{\pm 1628993}$ | $0.203_{\pm 0.017}$ | $88.547_{\pm 0.827}$ |
| | M-SASSHA | $129_{\pm 17}$ | $1.58_{\pm 0.08}$ | $1.551_{\pm 0.684}$ | $1.025_{\pm 0.36}$ | $0.112_{\pm 0.008}$ | $92.363_{\pm 0.23}$ |
| | SASSHA | $78_{\pm 5.09}$ | $0.86_{\pm 0.03}$ | $0.184_{\pm 0.053}$ | $0.388_{\pm 0.704}$ | $0.117_{\pm 0.003}$ | $92.983_{\pm 0.05}$ |
| ResNet32 | SGD | $56_{\pm 5.10}$ | $0.80_{\pm 0.04}$ | $0.560_{\pm 0.219}$ | $0.196_{\pm 0.146}$ | $0.299_{\pm 0.002}$ | $92.693_{\pm 0.06}$ |
| | SAM | $45_{\pm 2.67}$ | $0.58_{\pm 0.02}$ | $0.107_{\pm 0.005}$ | $0.753_{\pm 0.351}$ | $0.128_{\pm 0.001}$ | $93.893_{\pm 0.13}$ |
| | Sophia-H | $7167_{\pm 2755}$ | $18.82_{\pm 5.50}$ | $9.399_{\pm 2.283}$ | $7.915_{\pm 3.397}$ | $0.418_{\pm 0.007}$ | $91.985_{\pm 0.08}$ |
| | AdaHessian | $1746_{\pm 1018}$ | $17.06_{\pm 10.24}$ | $4.599_{\pm 1.71}$ | $5.518_{\pm 3.623}$ | $0.253_{\pm 0.006}$ | $92.483_{\pm 0.15}$ |
| | Shampoo | $717553_{\pm 93129}$ | $4523_{\pm 629.7}$ | $162.1_{\pm 123.2}$ | $105322_{\pm 82246}$ | $0.269_{\pm 0.005}$ | $90.227_{\pm 0.238}$ |
| | M-SASSHA | $283_{\pm 10}$ | $3.96_{\pm 0.10}$ | $2.986_{\pm 1.133}$ | $1.300_{\pm 0.969}$ | $0.081_{\pm 0.001}$ | $93.177_{\pm 0.30}$ |
| | SASSHA | $47_{\pm 1.88}$ | $0.59_{\pm 0.02}$ | $0.136_{\pm 0.019}$ | $0.714_{\pm 0.090}$ | $0.112_{\pm 0.001}$ | $94.093_{\pm 0.24}$ |
| CIFAR-100 | | | | | | | |
| ResNet32 | SGD | $265_{\pm 25}$ | $7.29_{\pm 0.30}$ | $0.703_{\pm 0.132}$ | $1.31_{\pm 1.03}$ | $1.027_{\pm 0.013}$ | $69.320_{\pm 0.19}$ |
| | SAM | $123_{\pm 11}$ | $2.63_{\pm 0.09}$ | $0.266_{\pm 0.025}$ | $-0.619_{\pm 0.594}$ | $0.512_{\pm 0.016}$ | $71.993_{\pm 0.20}$ |
| | Sophia-H | $22797_{\pm 10857}$ | $68.15_{\pm 20.19}$ | $8.13_{\pm 3.082}$ | $19.19_{\pm 6.38}$ | $1.251_{\pm 0.020}$ | $67.760_{\pm 0.37}$ |
| | AdaHessian | $11992_{\pm 5779}$ | $46.94_{\pm 17.60}$ | $4.119_{\pm 1.136}$ | $12.50_{\pm 6.08}$ | $0.982_{\pm 0.026}$ | $68.060_{\pm 0.22}$ |
| | Shampoo | $436374_{\pm 9017}$ | $6823.34_{\pm 664.65}$ | $73.27_{\pm 12.506}$ | $49307489_{\pm 56979794}$ | $0.508_{\pm 0.07}$ | $64.077_{\pm 0.46}$ |
| | M-SASSHA | $382_{\pm 65}$ | $8.75_{\pm 0.31}$ | $2.391_{\pm 0.425}$ | $2.26_{\pm 1.66}$ | $0.628_{\pm 0.010}$ | $70.93_{\pm 0.21}$ |
| | SASSHA | $107_{\pm 40}$ | $1.87_{\pm 0.65}$ | $0.238_{\pm 0.088}$ | $0.65_{\pm 0.86}$ | $0.425_{\pm 0.001}$ | $72.143_{\pm 0.16}$ |
| WRN28-10 | SGD | $18_{\pm 1.17}$ | $0.66_{\pm 0.04}$ | $1.984_{\pm 0.506}$ | $-0.007_{\pm 0.028}$ | $0.820_{\pm 0.005}$ | $80.063_{\pm 0.15}$ |
| | SAM | $9_{\pm 0.866}$ | $0.23_{\pm 0.01}$ | $0.841_{\pm 0.084}$ | $0.024_{\pm 0.041}$ | $0.648_{\pm 0.006}$ | $82.560_{\pm 0.13}$ |
| | Sophia-H | $3419_{\pm 3240}$ | $13.57_{\pm 3.30}$ | $5.073_{\pm 0.268}$ | $0.067_{\pm 0.054}$ | $0.864_{\pm 0.003}$ | $79.353_{\pm 0.24}$ |
| | AdaHessian | $35119_{\pm 46936}$ | $139.53_{\pm 190.98}$ | $6.745_{\pm 1.932}$ | $19.727_{\pm 27.866}$ | $1.005_{\pm 0.008}$ | $76.917_{\pm 0.26}$ |
| | Shampoo | $102129_{\pm 60722}$ | $1459.09_{\pm 709.42}$ | $483_{\pm 172}$ | $98.558_{\pm 123.082}$ | $1.168_{\pm 0.072}$ | $74.063_{\pm 1.279}$ |
| | M-SASSHA | $2257_{\pm 248}$ | $30.40_{\pm 4.78}$ | $4.599_{\pm 0.003}$ | $0.301_{\pm 0.047}$ | $0.729_{\pm 0.01}$ | $81.533_{\pm 0.27}$ |
| | SASSHA | $84_{\pm 3.15}$ | $2.03_{\pm 0.11}$ | $4.540_{\pm 0.122}$ | $0.007_{\pm 0.129}$ | $0.603_{\pm 0.002}$ | $83.543_{\pm 0.08}$ |

# B M-SASSHA: EFFICIENT PERTURBATION

Having explored techniques to reduce the computational cost of second-order methods, here we consider employing techniques to alleviate the additional gradient computation in sharpness-minimization. Prior works have suggested different ways to reduce this computational overhead including infrequent computations (Liu et al., 2022), use of sparse perturbations (Mi et al., 2022), or computing with selective weight and data (Du et al., 2022a). In particular, we employ the approaches of Becker et al. (2024), which uses the normalized negative momentum as the perturbation:

$$\epsilon_t^\star = \rho \frac{m_{t-1}}{\|m_{t-1}\|_2}, \tag{9}$$

which entirely eliminates the need for additional gradient computation with similar generalization improvement as the original SAM. We call this low-computation alternative as M-SASSHA and evaluate this alongside SASSHA in Section 5, which shows much similar performance at the cost of first-order methods like SGD or Adam.

# C ALGORITHM COMPARISON

In this section, we compare our algorithm with other adaptive and second-order optimizers designed for deep learning to better illustrate our contributions within concurrent literature. We present a detailed comparison of each optimizer in Table 8.

Adam (Kingma & Ba, 2015) is an adaptive optimizer popular among practitioners, which uses gradient momentum and the moving average of gradient second moment as a preconditioner inspired by Adagrad (Duchi et al., 2011) and RMSProp (Tieleman & Hinton, COURSERA: Neural networks

Table 8: Comparison of different optimization algorithms in terms of gradient moment $m_t$, diagonal preconditioner $D_t$, and other operations $\mathbf{U}(z)$ unique to each optimizer. Here $g_t$, $\widehat{H}_t$ are the stochastic gradient and the Hessian estimation respectively, and $\beta_1, \beta_2$ denotes the hyperparameters for gradient and preconditioner moment.

| | | | |
|---|---|---|---|
| $x_{t+1} = x_t - \eta_t \mathbf{U}(D_t^{-1} m_t)$ | | | |
| | $m_t$ | $D_t$ | $\mathbf{U}(z)$ |
| SGD with momentum | $\beta_1 m_{t-1} + (1-\beta_1)g_t$ | $I$ | $z$ |
| Stochastic Newton | $g_t$ | $H_t(x_t)$ | $z$ |
| Adam (Kingma & Ba, 2015) | $\beta_1 m_{t-1} + (1-\beta_1)g_t$ | $\sqrt{\beta_2 v_{t-1} + (1-\beta_2)\,\mathbf{diag(g_t g_t^\top)}}$ | $\mathtt{bc}(z)$ |
| AdamW (Loshchilov & Hutter, 2018) | " | " | $z + \lambda_t$ |
| AdaHessian (Yao et al., 2021) | " | $\sqrt{\beta_2 v_{t-1} + (1-\beta_2)\,\mathbf{\widehat{H}_t^{(s)}(x_t)^2}}$ | $\mathtt{bc}(z)$ |
| Sophia-H (Liu et al., 2024) | " | $\beta_2 v_{t-1} + (1-\beta_2)\,\mathbf{\widehat{H}_t^{(c)}(x_t)}$ every $k$ steps | $\mathbf{clip(z)}$ |
| SASSHA (Ours) | $\beta_1 m_{t-1} + (1-\beta_1)\,g_t(x_t+\epsilon_t^\star)$ | $\sqrt{\beta_2 v_{t-1} + (1-\beta_2)\,\mathbf{\lvert\widehat{H}_t(x_t+\epsilon_t^\star)\rvert}}$ every $k$ steps | $\mathtt{bc}(z)$ |

\* $\mathtt{bc}(\cdot)$: bias correction

for machine learning). This is closely related to second-order methods as they can be viewed as using a diagonal approximation of the Fisher information matrix with square root for more conservative adaptation to the geometry of the data. AdamW (Loshchilov & Hutter, 2018) propose to improve this by decoupling the weight decay from the Adam update for better generalization, which becomes a widely employed regularization strategy for second-order methods.

AdaHessian (Yao et al., 2021) is one of the initial attempts among recent efforts to design efficient second-order optimization for deep learning. As the name suggests, it draws many techniques from adaptive methods such as moving averages of second moments with bias corrections, and diagonal approximation to preconditioning. However, they also propose using techniques such as Hutchinson diagonal estimators (Hutchinson, 1989; Roosta-Khorasani & Ascher, 2014) and spatial averaging on the Hessian ($\widehat{H}_t^{(s)}$), which consists of averaging the diagonal element within a filter of a convolution layer for filter-wise gradient scaling. Sophia (Liu et al., 2024) is a stochastic second-order optimizer specifically designed for language model pretraining. Its primary feature is the use of the clipping mechanism $\mathtt{clip}(z) = \max\{\min\{z,\rho\},-\rho\}$ with a predefined threshold $\rho$ to control the negative impact of inaccurate Hessian estimations. Additionally, a hard adjustment is applied to each Hessian entry, substituting negative and very small values with a constant $\epsilon$, such as $\widehat{H}_t^{(c)} = \max\{\widehat{h}_t, \epsilon\}$ to pre-

---

**Algorithm 1** SASSHA and M-SASSHA

1: **Input:** Initial parameter $x_0$, learning rate $\{\eta_t\}$, moving average parameters $\beta_1, \beta_2$, Hessian update interval $k$, weight decay parameter $\lambda$
2: Set $m_{-1} = 0$, $D_{-1} = 0$
3: **for** $t = 0$ to $T$ **do**
4:    **if** SASSHA **then**
5:      $g_t = \nabla f_\mathcal{B}(x_t)$
6:      $\epsilon_t^\star = \rho g_t / \|g_t\|_2$
7:    **else if** M-SASSHA **then**
8:      $\epsilon_t^\star = \rho m_{t-1} / \|m_{t-1}\|_2$
9:    $\tilde{g}_t = \nabla f_\mathcal{B}(x_t + \epsilon_t^\star)$
10:   $m_t = \beta_1 m_{t-1} + (1-\beta_1)\tilde{g}_t$
11:   $\overline{m}_t = m_t / (1 - \beta_1^t)$
12:   **if** $t \bmod k = 0$ **then**
13:     $\hat{H}_t = \widehat{H}(x_t + \epsilon_t^\star)$
14:     $D_t = \beta_2 D_{t-1} + (1-\beta_2)|\tilde{H}_t|$
15:     $\overline{D}_t = \sqrt{D_t / (1 - \beta_2^t)}$
16:   **else**
17:     $\overline{D}_t = \overline{D}_{t-1}$
18:   $x_{t+1} = x_t - \eta_t \overline{D}_t^{-1} \overline{m}_t - \eta_t \lambda x_t$

vent convergence to saddle points and mitigate numerical instability. They also proposed using the Gauss-Newton-Bartlett diagonal estimator alongside the Hutchinson estimator. To further attain efficiency, they showed moderate robustness to lazy Hessian updates and proposed to update every 10 iterations of optimization, much longer compared to AdaHessian.

Our proposed method SASSHA adds additional perturbation $\epsilon_t^\star$ before computing the gradient and Hessian to penalize sharpness during the training process, which has not been explored in the literature. We find this sharpness minimization scheme also seems to aid lazy Hessian updates. This, however, can cause instability in the preconditioning, which we alleviate using square roots.

# D    CONVERGENCE ANALYSIS OF SASSHA

In this section, to ensure the completeness of our work, we provide preliminary convergence analysis results. Based on the well-established analyses of Li et al. (2023); Khanh et al. (2024), we further investigate the complexities arising from preconditioned perturbed gradients.

**Assumption D.1.** The function $f : \mathbb{R}^d \to \mathbb{R}$ is convex, $\beta$-smooth, and bounded from below, i.e., $f^* := \inf_x f(x) > -\infty$. Additionally, the gradient $\nabla f(x_t)$ is non-zero for a finite number of iterations, i.e., $\nabla f(x_t) \neq 0$ for all $t \in \{1, 2, \ldots, n\}$.

**Assumption D.2.** Step sizes $\eta_t$ and perturbation radii $\rho_t$ are assumed to satisfy the following conditions:

$$\sum_{t=1}^{\infty} \eta_t = \infty, \quad \sum_{t=1}^{\infty} \eta_t^2 < \infty, \quad \sum_{t=1}^{\infty} \rho_t^2 \eta_t < \infty.$$

*Remark* D.3. The following notations will be used throughout

1. $g_t := \nabla f(x_t)$ denotes the gradient of $f$ at iteration $t$.

2. The intermediate points and the difference between the gradients are defined as

$$x_{t+\frac{1}{2}} := x_t + \rho_t \frac{g_t}{\|g_t\|}, \quad g_{t+\frac{1}{2}} := \nabla f(x_{t+\frac{1}{2}}), \quad \delta_t := g_{t+\frac{1}{2}} - g_t.$$

3. For $u, v \in \mathbb{R}^d$, operations such as $\sqrt{v}, |v|$ and $\frac{v}{u}$, as well as the symbols $\preceq$ and $\succeq$, are applied element-wise.

*Remark* D.4. The update rule for the iterates is given by

$$x_{t+1} = x_t - \frac{\eta_t}{\sqrt{|\operatorname{diag}(\nabla^2 f(x_{t+\frac{1}{2}}))| + \epsilon}} \odot g_{t+\frac{1}{2}}, \tag{10}$$

where $\operatorname{diag}$ extracts the diagonal elements of a matrix as a vector, or constructs a diagonal matrix from a vector, and $\epsilon$ is a damping constant. Define $h_t$ as

$$h_t = \frac{\eta_t}{\sqrt{|\operatorname{diag}(\nabla^2 f(x_{t+\frac{1}{2}}))| + \epsilon}},$$

then the following hold

1. From the convexity and $\beta$-smoothness of $f$, the diagonal elements of $\nabla^2 f(x)$ are bounded within the interval $[0, \beta]$, i.e.,

$$0 \leq \left[\nabla^2 f(x)\right]_{(i,i)} = e_i^\top \nabla^2 f(x) e_i \leq \beta,$$

where $e_i$ is the $i$-th standard basis vector in $\mathbb{R}^d$.

2. The term $h_t$ is bounded as

$$\frac{\eta_t}{\sqrt{\beta} + \epsilon} \preceq h_t \preceq \frac{\eta_t}{\epsilon}.$$

*Remark* D.5. For the matrix representation

1. Denoting $H_t := \operatorname{diag}(h_t)$, the matrix bounds for $H_t$ are given by

$$\frac{\eta_t}{\sqrt{\beta} + \epsilon} I \preceq H_t \preceq \frac{\eta_t}{\epsilon} I, \tag{11}$$

where $I$ is the identity matrix.

2. Using the matrix notation $H_t$, the update for the iterates is expressed as

$$x_{t+1} = x_t - H_t g_{t+\frac{1}{2}}.$$

*Remark* D.6. From the $\beta$-smoothness of $f$, $\delta_t$ is bounded by

$$\|\delta_t\| \leq \beta \|x_t + \rho_t \frac{\nabla f(x_t)}{\|\nabla f(x_t)\|} - x_t\| = \beta \rho_t. \tag{12}$$

**Lemma D.7** (Descent Lemma). *Under Assumption D.1 and Assumption D.2, for given $\beta$ and $\epsilon$, there exists a $T \in \mathbb{N}$ such that for $\forall t \geq T$, $\eta_t$ satisfies $\eta_t \leq \min\left\{\frac{\epsilon^2}{6\beta(\sqrt{\beta}+\epsilon)}, \frac{\epsilon}{4\beta}\right\}$. For such $t \geq T$, the following inequality holds*

$$f(x_{t+1}) \leq f(x_t) - \frac{\eta_t}{2(\sqrt{\beta}+\epsilon)}\|g_t\|^2 + \frac{\eta_t}{\epsilon}\|\delta_t\|^2. \tag{13}$$

*Proof.* We begin by applying the $\beta$-smoothness of $f$,

$$f(x_{t+1}) \leq f(x_t) + \langle g_t, x_{t+1} - x_t \rangle + \frac{\beta}{2}\|x_{t+1} - x_t\|^2$$

$$= f(x_t) - \langle g_t, H_t(g_t + \delta_t)\rangle + \frac{\beta}{2}\|H_t(g_t + \delta_t)\|^2$$

$$\leq f(x_t) - g_t^\top H_t g_t + \frac{1}{2\alpha}g_t^\top H_t g_t + \frac{\alpha}{2}\delta_t^\top H_t \delta_t + \frac{\beta}{2}\|H_t(g_t + \delta_t)\|^2$$

$$\leq f(x_t) - (1 - \frac{1}{2\alpha})\frac{\eta_t}{\sqrt{\beta}+\epsilon}\|g_t\|^2 + \frac{\alpha}{2}\frac{\eta_t}{\epsilon}\|\delta_t\|^2 + \frac{\beta}{2}\frac{\eta_t^2}{\epsilon^2}\|g_t + \delta_t\|^2$$

$$\leq f(x_t) - (1 - \frac{1}{2\alpha})\frac{\eta_t}{\sqrt{\beta}+\epsilon}\|g_t\|^2 + \frac{\alpha}{2}\frac{\eta_t}{\epsilon}\|\delta_t\|^2 + \beta\frac{\eta_t^2}{\epsilon^2}(\|g_t\|^2 + \|\delta_t\|^2)$$

$$= f(x_t) - \eta_t((1 - \frac{1}{2\alpha})\frac{1}{\sqrt{\beta}+\epsilon} - \beta\frac{\eta_t}{\epsilon^2})\|g_t\|^2 + \eta_t(\frac{\alpha}{2\epsilon} + \beta\frac{\eta_t}{\epsilon^2})\|\delta_t\|^2.$$

The second inequality follows from Young's inequality, the third inequality is obtained from Equation (11), and the last inequality is simplified using the property $\|a + b\|^2 \leq 2\|a\|^2 + 2\|b\|^2$. By setting $\alpha = \frac{3}{2}$, we get

$$= f(x_t) - \eta_t(\frac{2}{3}\left(\frac{1}{\sqrt{\beta}+\epsilon}\right) - \beta\frac{\eta_t}{\epsilon^2})\|g_t\|^2 + \eta_t(\frac{3}{4\epsilon} + \beta\frac{\eta_t}{\epsilon^2})\|\delta_t\|^2.$$

Since $\eta_t \downarrow 0$, $\exists T \in \mathbb{N}$ such that $\eta_t \leq \min\{\frac{\epsilon^2}{6\beta(\sqrt{\beta}+\epsilon)}, \frac{\epsilon}{4\beta}\}$, this gives $\frac{2}{3}\left(\frac{1}{\sqrt{\beta}+\epsilon}\right) - \beta\frac{\eta_t}{\epsilon^2} \geq \frac{1}{2(\sqrt{\beta}+\epsilon)}$ and $\frac{3}{4\epsilon} + \beta\frac{\eta_t}{\epsilon^2} \leq \frac{1}{\epsilon}$, which implies

$$\leq f(x_t) - \frac{\eta_t}{2(\sqrt{\beta}+\epsilon)}\|g_t\|^2 + \frac{\eta_t}{\epsilon}\|\delta_t\|^2$$

$\square$

**Theorem D.8.** *Under Assumption D.1 and Assumption D.2, given any initial point $x_0 \in \mathbb{R}^d$, let $\{x_t\}$ be generated by Equation (10). Then, it holds that $\liminf_{t\to\infty}\|g_t\| = 0$.*

*Proof.* From Lemma D.7 and Equation (12), we have the bound

$$f(x_{t+1}) \leq f(x_t) - \frac{\eta_t}{2(\sqrt{\beta}+\epsilon)}\|g_t\|^2 + \frac{\eta_t}{\epsilon}\|\delta_t\|^2$$

$$\leq f(x_t) - \frac{\eta_t}{2(\sqrt{\beta}+\epsilon)}\|g_t\|^2 + \frac{\eta_t}{\epsilon}\beta^2\rho_t^2.$$

By rearranging the terms, we obtain the following

$$\frac{\eta_t}{2(\sqrt{\beta}+\epsilon)}\|g_t\|^2 \leq f(x_t) - f(x_{t+1}) + \frac{\eta_t}{\epsilon}\beta^2\rho_t^2.$$

For any $M > T$, we have

$$\frac{1}{2(\sqrt{\beta}+\epsilon)}\sum_{t=T}^{M}\eta_t\|g_t\|^2 \le \sum_{t=T}^{M}\left(f(x_t)-f(x_{t+1})\right)+\frac{\beta^2}{\epsilon}\sum_{t=T}^{M}\rho_t^2\eta_t$$

$$= f(x_T)-f(x_{M+1})+\frac{\beta^2}{\epsilon}\sum_{t=T}^{M}\rho_t^2\eta_t$$

$$\le f(x_T)-\inf_{t\in\mathbb{N}}f(x_t)+\frac{\beta^2}{\epsilon}\sum_{t=T}^{M}\rho_t^2\eta_t.$$

As $M \to \infty$, the series $\sum_{t=T}^{\infty}\eta_t\|g_t\|^2$ converges. Now, assume for contradiction that $\liminf_{t\to\infty}\|g_t\| \ne 0$. This means there exists some $\xi > 0$ and $N \ge T$ such that $\|g_t\| \ge \xi$ for all $t \ge N$. Consequently, we have

$$\infty > \sum_{t=N}^{\infty}\eta_t\|g_t\|^2 \ge \xi^2\sum_{t=N}^{\infty}\eta_t = \infty,$$

which is a contradiction. Therefore, $\liminf_{t\to\infty}\|g_t\| = 0$. $\qquad\square$

## E  EXPERIMENT SETTING

Here, we describe our experiment settings in detail.

**CIFAR**  We trained ResNet models on the CIFAR datasets for 160 epochs and Wide-ResNet28-10 for 200 epochs. We employed only standard inception-style data augmentations, such as random cropping and horizontal flipping, without any additional regularization techniques or data augmentations. The loss function used was cross-entropy. We utilized a multi-step decay learning rate schedule. Specifically, for ResNet20 and ResNet32, the learning rate was decayed by a factor of $0.1$ at epochs $60$ and $120$. For Wide-ResNet28-10, the learning rate was decayed by a factor of $0.2$ at epochs $80$ and $160$. The hyperparameters for exponential moving average were set to $\beta_1 = 0.9$ and $\beta_2 = 0.999$. A batch size of $256$ was used in all experiments. The hyperparameter search space for different optimizers is detailed in Table 9.

| Optimizer | SASSHA | M-SASSHA | AdaHessian | Sophia-H | AdamW / SGD | shampoo |
|---|---|---|---|---|---|---|
| Learning Rate | $\left\{0.3, 0.15\right\}$ | | $\left\{0.3, 0.15, 0.1, 0.03, 0.015, 0.01, 0.001\right\}$ | | | $\left\{\begin{smallmatrix}1.5,1.4,1.3,1.2,1.1,1,0.9,0.8,0.7,0.6,\\0.5,0.4,0.3,0.2,0.1,0.01,0.04,0.004\end{smallmatrix}\right\}$ |
| Weight Decay | | | $\left\{\text{2e-3, 1e-3, 5e-4, 1e-4, 5e-5, 1e-5, 5e-6, 1e-6}\right\}$ | | | |
| Perturbation radius $\rho$ | $\left\{0.15, 0.2, 0.25\right\}$ | $\left\{0.1, 0.2, 0.3, 0.6, 0.8\right\}$ | - | - | - | - |
| Clipping-threshold | - | - | - | $\left\{\begin{smallmatrix}0.1,0.05,0.01,0.005,\\0.001,0.0005,0.0001\end{smallmatrix}\right\}$ | - | - |
| Damping | - | - | - | - | - | 1e-$\left\{2, 3, 4, 6, 8\right\}$ |
| Hessian Update Interval $k$ | 10 | 10 | 1 | 1 | - | 1 |
| learning rate schedule | | | Multi-step decay | | | |

Table 9: Hyperparameter search space for CIFAR on ResNet

**ImageNet**  We trained ResNet50 and ViT-S/32 models on the ImageNet dataset for 90 epochs. Consistent with our CIFAR training settings, we utilized only standard inception-style data augmentations and employed the cross-entropy loss function. When training ResNet50, we used a multi-step decay learning rate schedule, reducing the learning rate by a factor of $0.1$ at epochs $30$ and $60$. However, for the AdaHessian, training was not possible with a multi-step decay schedule; therefore, following (Yao et al., 2021), we adopted a plateau decay schedule. For training the Vision Transformer model, following (Chen et al., 2022), we employed a cosine learning rate schedule with an 8-epoch warm-up phase. The $\beta_1$ and $\beta_2$ were set to $0.9$ and $0.999$ respectively. We used a batch size of $256$ for ResNet50 and a batch size of $1024$ for ViT. The hyperparameter search spaces for each optimizer used during training on the ImageNet dataset are detailed in Table 10.

| Optimizer | SASSHA | M-SASSHA | AdaHessian | Sophia-H | AdamW / SGD |
|---|---|---|---|---|---|
| Learning Rate | $\{0.3, 0.15\}$ | $\{0.3, 0.15\}$ | $\{0.3, 0.15\}$ | $\{0.1, 0.01, 0.001\}$ | |
| Weight Decay | | | $\{$2e-3, 1e-3, 5e-4, 1e-4, 5e-5, 1e-5$\}$ | | |
| Perturbation radius $\rho$ | $\{0.1, 0.15, 0.2, 0.25\}$ | $\{0.1, 0.2, 0.4, 0.8\}$ | - | - | - |
| Clipping-threshold | - | - | - | $\{0.1, 0.05, 0.01, 0.005, 0.001, 0.0005, 0.0001\}$ | - |
| Hessian Update Interval $k$ | 10 | 10 | 1 | 1 | - |

Table 10: Hyperparameter search space for ImageNet

**Language pretraining** Following the training settings introduced in (Gomes et al., 2024), we conducted experiments on a mini GPT-1 model using the Wikitext-2 dataset. This scaled-down version of GPT-1 instead of the original twelve, maintaining essential modeling capabilities while reducing computational demands. We trained the model with three optimizers: SASSHA, M-SASSHA, and Sophia-H. The hyperparameter tuning spaces for these optimizers are summarized in Table 11. For other optimizers not listed in the table, we directly reported the results from (Gomes et al., 2024).

| Optimizer | SASSHA / M-SASSHA | Sophia-H |
|---|---|---|
| Learning Rate | $\{0.15, 0.075, 0.015, 0.0075, 0.0015\}$ | $\{$1e-2, 5e-3, 1e-3, 5e-4, 1e-4, 5e-5, 1e-5$\}$ |
| Weight Decay | | 1e-$\{1, 2, 4, 6, 8\}$ |
| Perturbation radius $\rho$ | 2.5e-$\{1, 2, 3, 4\}$ | - |
| Clipping-threshold | - | $\{$1e-1, 5e-2, 1e-2, 5e-3, 1e-3, 5e-4, 1e-4$\}$ |
| Hessian Update Interval $k$ | 10 | 1 |
| Epochs | | 50 |

Table 11: Hyperparameter search space for Language pretraining

**Language Finetuning** In our experiments, we utilized a pretrained SqueezeBERT model from the HuggingFace Hub (Wolf et al., 2020) instead of pretraining the model from scratch as initially proposed by Iandola et al. (2020). For fine-tuning, we set the batch size to 16, the maximum sequence length to 512, and disabled dropout by setting the dropout rate to zero. The number of fine-tuning epochs varied according to the specific GLUE task: 5 epochs for MNLI, QQP, QNLI, and SST-2; 10 epochs for STS-B, MRPC, and RTE; and 20 epochs for CoLA. The detailed hyperparameter search spaces are presented in Table 12.

| Optimizer | SASSHA / M-SASSHA | Sophia-H | AdaHessian | AdamW |
|---|---|---|---|---|
| Learning Rate | 1e-$\{1, 2, 3\}$ | 1e-$\{1, 2, 3, 4, 5\}$ | 1e-$\{1, 2, 3, 4, 5\}$ | 1e-$\{1, 2, 3, 4, 5\}$ |
| Weight Decay | $\{$1e-4, 5e-5, 1e-5, 5e-6, 1e-6$\}$ | 1e-$\{4, 5, 6, 7, 8\}$ | 1e-$\{4, 5, 6, 7, 8\}$ | 1e-$\{4, 5, 6, 7, 8\}$ |
| Perturbation radius $\rho$ | 2.5e-$\{1, 2, 3, 4\}$ | - | - | - |
| Clipping-threshold | - | $\left\{\begin{array}{c}0.1, 0.05, 0.01, 0.005,\\ 0.001, 0.0005, 0.0001\end{array}\right\}$ | - | - |
| Hessian Update Interval $k$ | 1 | 1 | 1 | - |

Table 12: Hyperparameter search space for language finetuning

**Label noise** To evaluate the robustness of ResNet32 under varying levels of label noise, we conducted a comprehensive grid search on the CIFAR datasets. We trained the model using a multi-step decay learning rate schedule while introducing label noise at rates of 20%, 40%, and 60%. The specific hyperparameters explored during these experiments are detailed in in Table 13.

| Optimizer | SASSHA | M-SASSHA | Sophia-H | AdaHessian | SAM | AdamW |
|---|---|---|---|---|---|---|
| Learning Rate | $\{0.3, 0.15, 0.1, 0.05, 0.015, 0.01, 0.001\}$ | | | | | |
| Weight Decay | $\{\text{5e-4, 5e-5, 1e-5, 5e-6, 1e-6}\}$ | | | | | |
| Perturbation radius $\rho$ | $\{0.1, 0.15, 0.2, 0.25\}$ | $\{0.1, 0.2, 0.3, 0.6, 0.8\}$ | - | - | $\{0.05, 0.1, 0.15, 0.2, 0.25\}$ | - |
| Clipping-threshold | - | - | $\{\begin{smallmatrix}0.1, 0.05, 0.01, 0.005, \\ 0.001, 0.0005, 0.0001\end{smallmatrix}\}$ | - | - | - |
| Hessian Update Interval $k$ | 10 | 10 | 1 | 1 | - | - |

Table 13: Hyperparameter search space for label noise experiments

## F  VALIDATION LOSS CURVE FOR VISION TASK

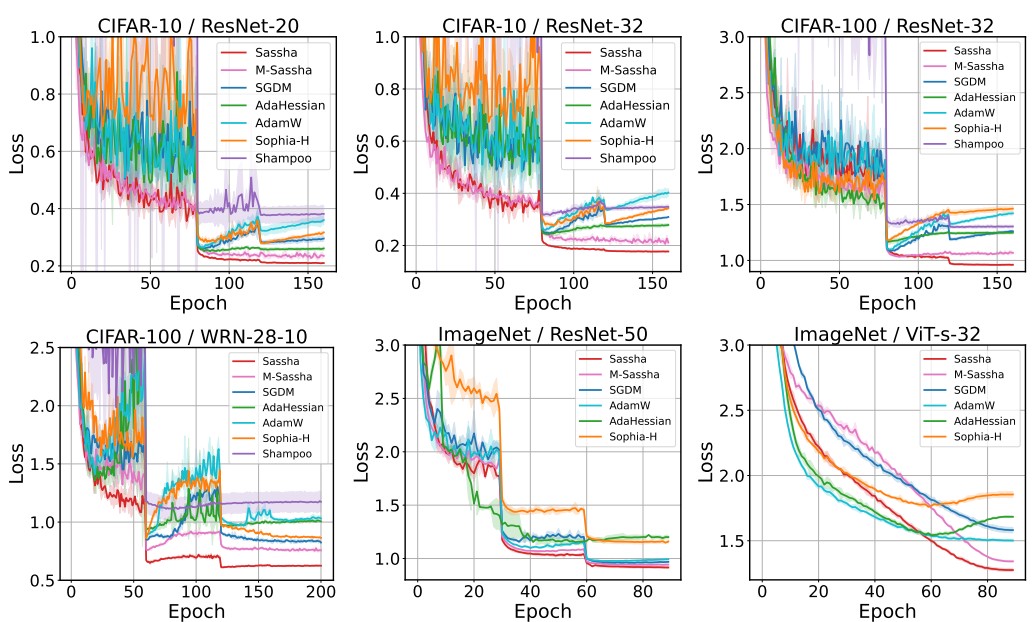

Figure 6: Validation loss curve of SASSHA, M-SASSHA, SGD, AdaHessian, AdamW, and Sophia-H on various image classification models and tasks. SASSHA outperforms all first-order and second-order baseline optimizers.

The experimental results demonstrate the better generalization capability of SASSHA over the related optimizers. Across all datasets and model architectures, our method consistently achieves the lowest validation loss, indicative of its enhanced ability to generalize from training to validation data effectively. This robust performance underscores SASSHA's potential as a leading optimization method for various deep learning applications, particularly in the domain of image classification.

# G ADDITIONAL ABLATION

## G.1 ABLATION OF THE ABSOLUTE FUNCTION

We observe how the absolute function influences the training process to avoid convergence to a critical solution that could result in sub-optimal performance. We train ResNet-32 on CIFAR-100 using SASSHA without the absolute function (No-Abs) and compare the resulting training loss to that of the original SASSHA. We also plot the Hessian eigenspectrum of the found solution via the Lanczos algorithm (Yao et al., 2020) to determine whether the found solution corresponds to a minimum or a saddle point. The results are illustrated in Figure 7. We can see that without the absolute function, the training loss converges to a sub-optimal solution, where the prevalent negative values in the diagonal Hessian distribution indicate it as a saddle point. This shows the necessity of the absolute function for preventing convergence to these critical regions.

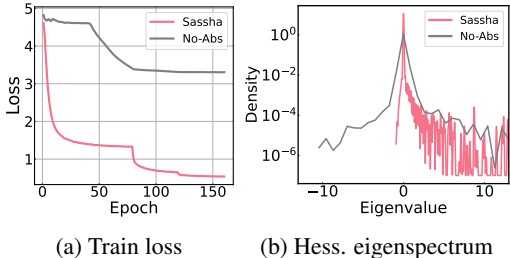

(a) Train loss  (b) Hess. eigenspectrum

Figure 7: Effect of the absolute function on the training loss and the Hessian eigenspectrum of the found solution of SASSHA on ResNet-32/CIFAR-10. Without the absolute function, SASSHA converges to sub-optimal saddle point.

## G.2 ADDITIONAL RESULTS FOR SASSHA VS SAM

Table 14: We conducted a comparative analysis of SASSHA and SAM across various datasets and models. The results indicate that SASSHA achieves a comparable level of validation accuracy in a shorter amount of time compared to SAM.

| | Epoch | Time (s) | Accuracy (%) |
|---|---|---|---|
| CIFAR10/RN20 | | | |
| SAM $_{\text{SGD}}$ | 160 | 956 | $92.847_{\pm 0.07}$ |
| SAM $_{\text{AdamW}}$ | 160 | 988 | $92.767_{\pm 0.29}$ |
| SASSHA | 120 | 936 | $\mathbf{92.873}_{\pm 0.05}$ |
| CIFAR10/RN32 | | | |
| SAM $_{\text{SGD}}$ | 160 | 1,466 | $93.893_{\pm 0.13}$ |
| SAM $_{\text{AdamW}}$ | 160 | 1,473 | $93.450_{\pm 0.24}$ |
| SASSHA | 120 | 1,440 | $\underline{93.810}_{\pm 0.12}$ |
| SASSHA | 160 | 1,920 | $\mathbf{94.093}_{\pm 0.24}$ |
| CIFAR100/RN32 | | | |
| SAM $_{\text{SGD}}$ | 160 | 1,471 | $71.993_{\pm 0.20}$ |
| SAM $_{\text{AdamW}}$ | 160 | 1,472 | $71.153_{\pm 0.37}$ |
| SASSHA | 120 | 1,447 | $\underline{71.920}_{\pm 0.30}$ |
| SASSHA | 160 | 1,930 | $\mathbf{72.143}_{\pm 0.16}$ |
| CIFAR100/WRN-28-10 | | | |
| SAM $_{\text{SGD}}$ | 200 | 23,692 | $83.036_{\pm 0.13}$ |
| SAM $_{\text{AdamW}}$ | 200 | 23,820 | $82.880_{\pm 0.31}$ |
| SASSHA | 150 | 21,309 | $\mathbf{83.167}_{\pm 0.15}$ |

### G.3 COST ANALYSIS

Table 15: Wall clock time (s) per epoch and the number of backward passes (BP) required for various optimizers, with the lazy Hessian interval $k$ of SASSHA set to 10. Both SASSHA, and M-SASSHA are significantly faster than other approximate second-order optimizers. Notably, M-SASSHA consistently demonstrates better speed than SAM across all settings.

| Optimizer | Avg. BP (theoretical) | CIFAR10 | | CIFAR100 | | ImageNet | |
|---|---|---|---|---|---|---|---|
| | | ResNet20 | ResNet32 | ResNet32 | WRN28-10 | ResNet50 | ViT-small |
| AdamW | 1 BP | 3.35 | 5.03 | 5 | 59.29 | 1356.36 | 976.56 |
| SAM | 2 BP | 5.97 | 9.16 | 9.19 | 118.46 | 3003.00 | 1302.08 |
| Sophia-H | 2 BP | 21.20 | 33.90 | 33.87 | 295.31 | 12512.50 | 2152.19 |
| AdaHessian | 2 BP | 20.10 | 33.75 | 31.64 | 296.63 | 12262.25 | 2077.07 |
| SASSHA | 2.1 BP | 7.80 | 12.00 | 12.06 | 142.06 | 3503.50 | 1377.20 |
| M-SASSHA | 1.1 BP | 5.70 | 8.91 | 8.89 | 84.12 | 2497.50 | 1065.40 |

Here we discuss the theoretical computation cost of SASSHA and M-SASSHA in terms of backpropagation query. The average backpropagation cost (*i.e.*, total BP / number of iterations) of SASSHA is $(2 + 1/k)$ BP. For the lazy Hessian interval $k = 10$ used in our evaluations, this corresponds to 2.1 BP. The calculation is as follows: when performing a total of $T$ iterations, the total cost includes $T$ BP for gradient calculation, $T$ BP for sharpness minimization, and $T/k$ BP for diagonal Hessian approximation performed once every $k$ iterations. This results in a total of $(2 + 1/k)T$ BP, yielding an average of $(2 + 1/k)$ BP per iteration. Compared with SAM, SASSHA requires only 5% more BP on average. M-SASSHA significantly reduces the cost, only requiring 10% BP compared to Adam/SGD. To measure these resource consumption in practice, we report the wall-clock time of various optimizers in Table 15 which shows that both SASSHA, and M-SASSHA are significantly faster than other approximate second-order optimizers. Notably, M-SASSHA consistently demonstrates better speed than SAM across all settings.

## H COMPUTING RESOURCES

The computations for this research were performed on a GPU cluster featuring nodes equipped with the following GPU resources:

- NVIDIA GeForce RTX 3090 GPUs, each with 24 GB of memory.
- NVIDIA A100 GPUs, each with 80 GB of memory.
- NVIDIA RTX A6000 GPUs, each with 48 GB of memory

The software stack used includes a Linux operating system, Slurm for resource management, and essential libraries such as CUDA and cuDNN. This setup provided the necessary computational power and efficiency to perform the extensive simulations and data processing required for this research.

