# OpenReview forum: "Sassha: Sharpness-aware Adaptive Second-order Optimization with Stable Hessian Approximation"
_ICLR.cc/2025/Conference — Submitted to ICLR 2025_

### Official Review · Reviewer_yeu1 · 2024-11-02

**Soundness:** 2
**Presentation:** 3
**Contribution:** 2
**Rating:** 5
**Confidence:** 2

**Summary:**

This paper addresses the generalization limitation of approximate second-order optimization methods. An approach called 'SASSHA' is proposed to enhance generalization by explicitly reducing sharpness. Empirical experiments are conducted to validate the effectiveness and robustness of SASSHA. It is demonstrated that SASSHA achieves good performance and strong generalization in noisy data settings and outperforms other methods.

**Strengths:**

1. Detailed explanation of techniques used in ‘SASSHA’ is given.

2. A series of experiments are conducted to verify the improvement of ‘SASSHA’.

**Weaknesses:**

1. See the question part.

2. Some incomplete references in line 113.

**Questions:**

1. In Section 5, empirical experiments compare the performance of SASSHA with some baselines. How does the empirical performance of ‘SASSHA’ compare with those approximate second-order algorithms similar to SASSHA, particularly those mentioned in Section 4.1& 4.2?  Only one comparison is presented in Section 5.4.

---

> ### Author Response · Authors · 2024-11-21
> **Response to yeu1**
>
> We really appreciate the reviewer’s feedback. While we address the reviewer’s specific comments below, we would be keen to engage in any further discussion.
>
> ---
>
> **More comparisons?**
>
> It appears that there may be a misunderstanding. All approximate second-order algorithms cited in Sections 4.1 and 4.2 (i.e. AdaHessian [1], Sophia [2], and Shampoo [3]) are compared in Section 5. Other algorithms cited in Sections 4.1 and 4.2 [5-11] are not approximate second-order methods and are thus excluded from the comparison (except SAM [4]). We note that for SAM [4] in Section 5.4, additional comparisons are presented in Section 6.3 and Appendix G.2. If this does not address your concerns or if there are any other comparisons you would like us to make, please let us know. We would be keen to provide them during the discussion period.
>
> ---
>
> **Errata**
>
> > Some incomplete references in line 113.
>
> Thank you for bringing this to our attention. We will have them fixed in the revised version.
>
> ---
>
> **Remark**
>
> We thank the reviewer for taking the time to review our work. Please let us know if there are any remaining questions or suggestions that could help improve the quality of our paper.
>
> &nbsp;
>
> **Reference**\
> [1] Yao et al., Adahessian: An adaptive second order optimizer for machine learning. AAAI, 2021.\
> [2] Liu et al., Sophia: A scalable stochastic second-order optimizer for language model pre-training. ICLR, 2024.\
> [3] Gupta et al., Shampoo: Preconditioned stochastic tensor optimization. ICLR, 2018.\
> [4] Foret et al., Sharpness-aware minimization for efficiently improving generalization. ICLR, 2021. \
> [5] Chaudhari et al.,  Entropy-SGD: Biasing gradient descent into wide valleys. ICLR, 2017. \
> [6] Izmailov et al., 2018 Averaging weights leads to wider optima and better generalization. UAI, 2018. \
> [7] Orvieto et al., Anticorrelated noise injection for improved generalization. ICML, 2022. \
> [8] Levenberg et al., A method for the solution of certain non-linear problems in least squares. QAM, 1944. \
> [9] Marquardt et al., An algorithm for least-squares estimation of nonlinear parameters. SIAM, 1963. \
> [10] Amari et al., adaptive method of realizing natural gradient learning for multilayer perceptrons. Neural computation, 2000.\
> [11] Nesterov & Polyak et al., cubic regularization of newton method and its global performance. Math. prog., 2006.

---

> > ### Author Response · Authors · 2024-11-27
> > **Gentle reminder**
> >
> > Dear Reviewer,
> >
> > We sincerely thank you for your time and valuable feedback. Given that some time has passed since our initial response, we would greatly appreciate it if you could review our response. We would be keen to address any other remaining concerns.
> >
> > Best wishes,\
> > The authors.

---

> > ### Comment · Reviewer_yeu1 · 2024-11-30
> >
> > Thanks for your response and for fixing the incomplete references. However, I am afraid there still seems to be a misunderstanding of the original question. Question 1 asks whether it is possible to compare the performance of ‘sassha’ with those algorithms (including but not limited to approximate second-order algorithms) in Section 4.1&4.2 that are related to the design of ‘sassha’ and its idea to some extent. The comparisons of experiment results in approximate second-order algorithms [1,2,3] are also not limited to approximate second-order algorithms. It appears that the authors also agree that not only approximate second-order algorithms need to be included in the experiments comparison, but also other related algorithms in the experiments comparison. Therefore, excluding these algorithms from the comparison because they are not approximate second-order algorithms seems not to be a convincing reason.
> >
> > [1] Yao et al., Adahessian: An adaptive second order optimizer for machine learning. AAAI, 2021.
> >
> > [2] Liu et al., Sophia: A scalable stochastic second-order optimizer for language model pre-training. ICLR, 2024.
> >
> > [3] Gupta et al., Shampoo: Preconditioned stochastic tensor optimization. ICLR, 2018.

---

> ### Author Response · Authors · 2024-12-03
> **Additional response**
>
> > whether it is possible to compare the performance of ‘sassha’ with those algorithms in Section 4.1&4.2 that are related to the design of ‘sassha’ and its idea to some extent.
>
> First, we list up all the referenced works in Sections 4.1 and 4.2 here and describe in detail why they are excluded for comparisons:
>
> `Section 4.1 [1-3]` : other sharpness minimization algorithms
> - Entropy-SGD [1] demands several additional backpropagations per update to minimize local entropy, which incurs impractical costs in deep learning while offering only marginal improvements over SGD.
> - SWA [2] is an algorithm that averages weight points to find flat solutions, requiring keeping several parameters and computing multiple gradients in each iteration. It is also known to be less effective compared to SAM [4].
> - Anti-PGD [3] proposes to inject a special type of noise in the weights to artificially mimic the noise of SGD, which is not as effective as SAM.
>
> `Section 4.2 [5-8]` : classical second-order techniques
> - Levenberg-Marquardt [5-6] requires resource-intensive computing and storing the inverse of the Gauss-Newton matrix, tracking changes in the loss function, and demanding additional matrix calculations to adaptively adjust its damping factor.
> - Natural Gradient Descent [7] similarly demands the computation and storage of the inverse of the Fisher information matrix for preconditioning.
> - Cubic Regularization [8] involves computing the Hessian matrix during optimization, repeatedly calculating Hessian-vector products, and solving complex subproblems.
>
> In short, we chose not to compare the sharpness-minimization algorithms [1-3] for two key reasons: (i) their mechanisms differ fundamentally from Sassha's and are **not directly related**; and (ii) they are **more resource-intensive and less competitive** than SAM, making SAM a sufficient baseline for demonstrating Sassha's advantages. Similarly, the classical second-order algorithms [5-8] were excluded due to their **computational infeasibility in deep learning**, requiring $\mathcal{O(d^3)}$ for computation and $\mathcal{O(d^2)}$ for memory.
>
> We emphasize that the algorithms selected for comparison in this paper were carefully chosen to ensure relevance, scalability, and objectivity. The exclusion of certain algorithms from our comparison was not simply because they are not approximate second-order algorithms, but was aimed at providing a focused and cohesive evaluation.
>
> ---
>
> **More comparison**
>
> Nevertheless, we additionally conducted further experiments on algorithms related to Sassha:
>
> The square root technique used in Sassha for stability improvement is related to damping and clipping. We also conducted experiments comparing the square root with these two techniques.
>
> |Optimizer|RN32-CIFAR100|
> |-|-|
> |damping |71.27|
> |clipping |69.47|
> |Sassha |**72.113**|
>
> We provide additional experiments comparing Sassha with SAM on Language tasks.
>
> `Pretraining`
>
> ||test ppl|
> |-|-|
> |SAM(AdamW)|158.06|
> |SASSHA|**122.40**|​
>
> `Finetuning`
>
> |Task|RTE|STSB|MRPC|SST2|QQP|MNLI|QNLI|
> |-|-|-|-|-|-|-|-|
> |Metric|Acc|Spearman/Pearson|Acc/F1|Acc|Acc/F1|Mat / M.Mat|Acc|
> |SAM(AdamW)|72.56|88.98 / 89.31|84.56 / 88.89|90.83|90.124 / 86.785|**81.79 / 82.65**|89.07|
> |Sassha|**73.29**|**89.29 / 89.66**|**87.75 / 91.23**|**91.51**|**91.00 / 87.93**|81.74 / 82.12|**89.86**|
>
>
> In short, Sassha was carefully designed to become a practical approximate second-order method and strong alternative to existing methods as demonstrated by these comparisons.
>
> If there is any other comparison that the reviewer believes should be included, we kindly request the reviewer for specific pointers to it along with rationale. We would be keen to provide them in the final version, if not during the discussion period. Otherwise, we would greatly appreciate it if the reviewer could reconsider the earlier critique and kindly reflect that on the rating of our work.
>
> &nbsp;
>
> **Reference**\
> [1] Chaudhari et al.,  Entropy-SGD: Biasing gradient descent into wide valleys. ICLR, 2017. \
> [2] Izmailov et al., 2018 Averaging weights leads to wider optima and better generalization. UAI, 2018. \
> [3] Orvieto et al., Anticorrelated noise injection for improved generalization. ICML, 2022. \
> [4] Jean Kaddour et al., When Do Flat Minima Optimizers Work?. Neurips, 2022 \
> [5] Levenberg et al., A method for the solution of certain non-linear problems in least squares. QAM, 1944. \
> [6] Marquardt et al., An algorithm for least-squares estimation of nonlinear parameters. SIAM, 1963. \
> [7] Amari et al., adaptive method of realizing natural gradient learning for multilayer perceptrons. Neural computation, 2000.\
> [8] Nesterov & Polyak et al., cubic regularization of newton method and its global performance. Math. prog., 2006.

---

> > ### Author Response · Authors · 2024-12-04
> > **Closing remark**
> >
> > As the end of the discussion period is closing, we express our sincere gratitude for the time and effort you have dedicated to reviewing our work. We will ensure that our discussions are incorporated in the final manuscript. Also, we hope that we have adequately addressed most of your concerns. If so, we would greatly appreciate it if the reviewer could give a re-consideration to the rating of this work.

---

### Official Review · Reviewer_kwCD · 2024-11-03

**Soundness:** 2
**Presentation:** 3
**Contribution:** 2
**Rating:** 6
**Confidence:** 4

**Summary:**

This paper targets the issue of poor generalization in second-order deep learning optimization methods, which has been a significant barrier to wider application despite their theoretical advantages. This work first provides empirical evidence that existing second-order methods converge to sharper minima compared to SGD, potentially explaining their inferior generalization. The authors thus propose SASSHA, which combines sharpness-aware minimization with efficient second-order optimization, incorporating several technical innovations for stability and efficiency. The method is evaluated on computer vision and language tasks, showcasing consistent improvements over both first-order and second-order baselines. The theoretical analysis, while preliminary, provides useful insights into the convergence properties. The experimental results support the main claims, showing improved generalization, computational efficiency, and robustness to label noise.

**Strengths:**

**(S1) Clear and reasonable motivation:** The paper attempts to bridge second-order optimization with sharpness awareness, a direction that holds some merit. The systematic investigation connecting second-order optimization with solution sharpness is insightful, supported by comprehensive empirical evidence using multiple complementary metrics (eigenvalue analysis, loss perturbation, trace calculations).

**(S2) Technical Soundness:** The integration of sharpness awareness into second-order optimization is done thoughtfully, with each component carefully designed and theoretically justified. The square root pre-conditioner is a clear and insightful design that effectively addresses the numerical instabilities inherent in second-order methods while maintaining computational efficiency. The lazy Hessian update scheme demonstrates favorable trade-offs in second-order optimization, providing significant computational benefits without sacrificing performance.

**(S3) The Presentation Clarity:** This manuscript demonstrates reasonable organization and writing clarity that makes its technical content accessible. The progression from motivation through empirical observation of second-order methods' convergence to sharp minima to the proposed SASSHA method, follows a logical flow. The experiments are presented in a systematic manner with appropriate tables and figures, particularly the visualization of loss landscapes in Figure 2 which effectively shows the sharpness differences between optimizers. While the mathematical notation is mostly consistent and the algorithmic description is complete, the authors could have provided more intuitive explanations of key concepts, especially regarding the interaction between sharpness awareness and second-order information.

**Weaknesses:**

**(W1) Technical Originality:** However, the core idea of combining sharpness-awareness with second-order information is relatively straightforward and could be considered incremental. The empirical investigation of sharpness measures largely confirms known intuitions about the relationship between curvature and generalization. The technical contribution, while potentially useful and providing knowledge advancement to the optimization community, builds directly on existing work (SAM and diagonal Hessian approximation) with limited fundamental breakthroughs.

**(W2) Critical Experimental Gaps:** The experiments suffer from significant oversights that cast doubt on the method's practical applications. The authors avoid testing on large-scale models (>100M parameters), raising questions about the scalability to modern deep neural networks. Furthermore, the absence of results on fundamental computer vision tasks, such as object detection and semantic segmentation,  suggests potential limitations in the performance consistency of the method. More importantly, the paper lacks comparison with recent sharpness-aware variants like ASAM [1] and GSAM [2], making it impossible to assess whether the proposed method represents genuine progress in the field.

**(W3) Technical Concerns:** The square root pre-conditioner, while empirically shown to provide stability benefits, appears to be an arbitrary choice without a rigorous mathematical foundation. The authors fail to explore or justify why this specific power transformation is optimal, neglecting to investigate other potential functional forms that might provide superior conditioning. The hyper-parameter k for Hessian update intervals introduces additional complexity to the already challenging problem of tuning optimization parameters, yet the paper provides insufficient guidance on its selection across different architectures and tasks. In addition, the memory requirements of storing and updating the diagonal Hessian approximation, though lower than full second-order optimizers, could still be prohibitive for large-scale applications, especially in resource-constrained environments. I recommend the authors provide additional experiments and discussions on these aspects.


**(W4) Implementation Complexity:** The proposed SASSHA method requires maintaining multiple sets of statistics and careful coordination between the sharpness-aware perturbation and Hessian approximation components. The potential for numerical instability in regions of low curvature is particularly worrying, as the paper does not provide robust safeguards against such scenarios. The absence of adaptive strategies for crucial hyper-parameters like the perturbation radius means practitioners must rely on costly trial-and-error tuning.

---

### Reference

[1] Asam: Adaptive sharpness-aware minimization for scale-invariant learning of deep neural networks. ICML, 2021

[2] Surrogate gap minimization improves sharpness-aware training. ICLR, 2021

**Questions:**

Please refer to Weaknesses for detailed questions. I hope my review helps to further strengthen this paper and helps the authors, fellow reviewers, and Area Chairs understand the basis of my recommendation. I also look forward to the rebuttal feedback and further discussions, and would be glad to raise my rating if thoughtful responses and improvements are provided.


## **-------------------- Post-Rebuttal Summary --------------------**

The additional experiments, discussions, and revised manuscript provided by the authors have significantly strengthened the work and addressed most of my concerns. I suppose this work now can provide knowledge advancement to the community, and I look forward to the final version manuscript, which incorporates the additional insights and information presented in the rebuttal stage.

---

> ### Author Response · Authors · 2024-11-24
> **Response to kwCD (1/3)**
>
> We are greatly encouraged by the reviewer’s positive and constructive feedback, which we believe will tremendously improve our work. While we address specific comments below, we would be keen to engage in any further discussion.
>
> ---
>
> **W1: Technical originality?**
>
> > “combining sharpness-awareness with second-order information is relatively straightforward and could be considered incremental”
>
> With all due respect, we disagree with the reviewer’s criticism. Simply put, this “combining” process was far from trivial. Achieving seamless integration of sharpness minimization into second-order optimization, **while retaining their benefits and avoiding instability and inefficiency**, indeed required extra careful engineering. This process involved extensive verifications of a diverse set of algorithmic design choices.
>
> It is also worth noting that such engineering is a standard procedure in optimization. Seemingly simple but well-thought-through ideas often result in significant performance improvement and have been appreciated by the community. For example, Clipping is the core component that distinguishes Sophia from other approximate second-order optimizers [1]. Similarly, AdamW’s decoupling of weight decay from L2 regularization is a widely appreciated innovation despite its simplicity [2].
>
> > “The empirical investigation of sharpness measures largely confirms known intuitions about the relationship between curvature and generalization.”
>
> We also find this criticism hard to accept, as to the best of our knowledge, no prior work has explicitly investigated nor confirmed what we show in this work (i.e., the correlation between the poor generalization of approximate second order methods and the sharpness of their solution) to any degree matching ours.
>
> Based on these points, we hope the reviewer could reconsider the value of our contributions.
>
> ---
>
> **W2: Critical experimental gaps?**
>
> > The authors avoid testing on large-scale models (>100M parameters), raising questions about the scalability to modern deep neural networks.
>
>
> We test Sassha on the fine-tuning task using the BERT Base model (110M parameters). The results are shown below:
>
> |Task|RTE|STSB|MRPC|SST2|QQP|MNLI|QNLI|
> |-|-|-|-|-|-|-|-|
> |Metric|Acc|Spearman/Pearson|Acc/F1|Acc|Acc/F1|Mat / M.Mat|Acc|
> |Sassha|**71.48**|89.29/89.71|**87.25**/**90.97**|**92.66**|91.11/88.01|**84.11**/**85.05**|90.96|
> |Sam(AdamW)|69.31|89.47/**89.85**|85.05/89.82|92.43|**91.14**/**88.05**|83.71/83.82|**91.07**|
> |adamW|68.23|**89.94**/89.34|85.05/89.78|91.74|90.84/87.58|83.25/83.56|90.55|
> |Sophia-H|68.95|12.38/12.38|87.01/90.69|82.22|77.92/66.76|58.07/56.703|61.413|
>
> This indicates that Sassha achieves better performance than others in most tasks. We will include this in the updated paper.
>
>
> > the absence of results on fundamental computer vision tasks, such as object detection and semantic segmentation, suggests potential limitations in the performance consistency of the method
>
> While it is understandable to be curious about the full potential, we find this request quite out of context and beyond the scope of this work.
>
> > More importantly, the paper lacks comparison with recent sharpness-aware variants like ASAM [1] and GSAM [2], making it impossible to assess whether the proposed method represents genuine progress in the field.
>
> We perform experiments to compare Sassha against advanced SAM variants [4-5]. For a fair comparison, we also evaluate G-Sassha (Sassha with surrogate gap-guided sharpness from GSAM[5]). The results are below.
>
> ||ASAM|GSAM|Sassha|G-Sassha |
> |-|-|-|-|-|
> |RN20-CIFAR10|92.96%|92.71%|**92.983%**|92.943%|
> |RN32-CIFAR 10|93.62%|93.76%|94.093%|**94.15%**|
> |RN32-CIFAR 100|71.6%|72.10%|72.143%|**72.18%**|
> |RN50-ImageNet|76.404%|**76.450%**|76.429%|*|
> |ViT_s-ImageNet|68.26%|69.600%|69.195%|**69.673%**|
>
> We find that the performance of Sassha is comparable to, and mostly better than, these SAM variants. In addition, we note clearly that GSAM requires tuning additional hyperparameters (which are known to be quite important but sensitive). ASAM, according to Kwon et al., may need a wider $\rho$ search range than the original SAM, which implies a more involved search process.

---

> ### Author Response · Authors · 2024-11-24
> **Response to kwCD (2/3)**
>
> **W3: Technical concerns?**
>
> > The square root pre-conditioner, while empirically shown to provide stability benefits, appears to be an arbitrary choice without a rigorous mathematical foundation
>
>
> We thank the reviewer’s constructive criticism. Using square-rooted preconditioner can be viewed as geometric interpolation between the first order method $H^{-\alpha}$, which has been demonstrated to enable selecting an optimal preconditioner that balances the bias and the variance of the population risk, thereby minimizing generalization error [6]. In general, $\alpha = 1/2$ (i.e., square root) has shown moderate performance across various scenarios [6, 7, 8]. Additionally, square rooting can alleviate specific instabilities that approximate second-order optimizers suffer near minima. Specifically, since we posit that the commonly used diagonal Hessian estimators can scale inversely quadratic to the gradient, the preconditioned gradient update can scale inverse to the gradient [9, 10]. This behavior can lead to instability as the gradient scale diminishes as the model reaches near minima. Square-rooting the preconditioner can help mitigate such instabilities. We will add this discussion to improve our manuscript.
>
>
> > The hyper-parameter k for Hessian update intervals introduces additional complexity
>
> Sassha remains robust even with a very large $k$ as demonstrated in Fig 5 (a) in Section 6.2.. This is potentially due to the sharpness minimization process contributing to Hessians less frequently changing, thereby enabling reuse. We refer the reviewer to Section 5.4 where we provide an ablation analysis on this aspect. While we may not be able to precisely point to an “optimal” $k$, we can at least say that a reasonable value seems to suffice.
>
> > the memory requirements of storing and updating the diagonal Hessian approximation, though lower than full second-order optimizers, could still be prohibitive for large-scale applications
>
> The memory requirement is not prohibitive. Storing the diagonal Hessian requires the same amount of memory as storing second moments in Adam. Also, while updating the diagonal Hessian using Hessian-vector products (HVP) incurs additional memory overhead compared to gradient computation, [11] indicates that this is acceptable for most large-scale applications.
>
> Nonetheless, we also plan to provide efficient implementations that leverage HVP-free Hessian approximations such as the diagonal of the Gauss-Newton matrix, which only introduces a 5% increase in memory cost compared to Adam while preserving similar performance [1].

---

> > ### Author Response · Authors · 2024-11-24
> > **Response to kwCD (3/3)**
> >
> > **W4: Implementation Complexity**
> >
> > > SASSHA method requires maintaining multiple sets of statistics and careful coordination between the sharpness-aware perturbation and Hessian approximation components
> >
> > Sassha, with a careful design, has a similar tuning complexity to SAM with AdamW as the base optimizer and is no more sensitive than SAM variants or adaptive second-order optimization methods. Like SAM, Sassha involves tuning parameters such as learning rate, weight decay, and $\rho$ and $\rho$ within a comparable search range. In contrast, ASAM demands a larger $\rho$ search range than the original SAM [4], GSAM introduces an additional hyperparameter $\alpha$ that requires careful tuning [5], and Sophia may demand sophisticated tuning to the clipping threshold and the $\epsilon$ used to substitute negative or very small Hessian entries [1].
> >
> > > the paper does not provide robust safeguards against such scenarios
> >
> > Square-rooting efficiently addresses numerical instabilities caused by underestimation and surpasses robust but adhoc alternatives like damping (adding a small scalar value to Hessian) and clipping (max{h_t, \epsilon}). To demonstrate this, we conducted an ablation study comparing these three techniques under the same settings as Figure 4. We evaluate the distribution of diagonal Hessian entries (focusing on the range for the bottom 5% (min~5%ile)) at iterations 100, 150, and 200, and the validation accuracy (below **Only-SM** $=\text{Sassha} - \text{Sqrt}$, same as ‘No-sqrt’ in Figure 4).
> >
> > |iteration|100|150|200|
> > |-|-|-|-|
> > |Only-SM|7.007e-10/8.885e-06|6.972e-10/8.903e-06|6.937e-10/9.641e-06|
> > |damping|1.083e-04/1.906e-04|1.109e-04/ 2e-04|1.133e-04/2.062e-04|
> > |clipping|1e-04/1e-04|1e-04/1e-04|1e-04/1e-04|
> > |Sassha (sqrt)|1.59e-05/9.99e-05|2.03e-05/1.20e-04|2.46e-05/1.38e-04|
> >
> > |Optimizer|RN32-CIFAR100|
> > |-|-|
> > |damping|71.27|
> > |clipping|69.47|
> > |Sassha (sqrt)|**72.113**|
> >
> > Results show that all three techniques reduce numerical instabilities; however, damping and clipping underperform compared to Sassha with the square rooted preconditioner. We attribute this performance gap to the rigid transformations used in damping and clipping—either shifting Hessian entries entirely or abruptly replacing specific entries with a constant $\epsilon$. Without careful tuning, these transformations can result in incorrect Newton updates. In contrast, square-rooting offers smoother adjustments to Hessian entries, enabling better overall performance.
> >
> >
> > &nbsp;
> >
> > **Reference**\
> > [1] Liu et al., Sophia: A Scalable Stochastic Second-order Optimizer for Language Model Pre-training, ICLR, 2024. \
> > [2] Loshchilov et al., Decoupled Weight Decay Regularization, ICLR, 2019. \
> > [3] Foret et al., Sharpness-Aware Minimization for Efficiently Improving Generalization, ICLR, 2021. \
> > [4] Kwon et al., ASAM: Adaptive Sharpness-Aware Minimization for Scale-Invariant Learning of Deep Neural Networks. ICML, 2021. \
> > [5] Zhuang et al., Surrogate Gap Minimization Improves Sharpness-Aware Training. ICLR, 2022. \
> > [6] Amari et al., When does Preconditioning Help or Hurt Generalization? ICLR, 2021. \
> > [7] Duchi et al., Adaptive Subgradient Methods for Online Learning and Stochastic Optimization. JMLR, 2011. \
> > [8] Kingma & Ba et al., Adam: A Method for Stochastic Optimization. ICLR, 2015. \
> > [9] Kunstner et al., Limitations of the Empirical Fisher Approximation for Natural Gradient Descent. NeurIPS, 2019. \
> > [10] Pascanu et al., Revisiting natural gradient for deep networks. ICLR, 2014.\
> > [11] How to compute Hessian-vector products?, ICLR blog post, 2024

---

> > > ### Author Response · Authors · 2024-11-29
> > > **Awaiting your response**
> > >
> > > Dear Reviewer,
> > >
> > > We want to thank you again for your time and thoughtful feedback on our submission. Given that some time has passed since we shared our responses, we would greatly appreciate it if you could review our rebuttal, check whether our response addresses the concerns, and re-evaluate the score accordingly. Please let us know if there is anything else you want us to explain more. We are eager to engage in further discussions and address any remaining concerns.
> > >
> > > Best wishes,\
> > > The authors.

---

> > > > ### Comment · Reviewer_kwCD · 2024-11-30
> > > > **Official Response from Reviewer kwCD to the Rebuttal**
> > > >
> > > > Dear Authors of Submission 2572,
> > > >
> > > > I have thoroughly reviewed the authors' responses and carefully examined all the additional experimental results provided in the rebuttal. The authors have effectively addressed most of my initial concerns through extra experiments and detailed discussions. After careful consideration of all the new evidence, I decide to raise my rating to 6 for the following reasons:
> > > >
> > > > **(1) Regarding Larger-scale Validation:**
> > > >
> > > > The new experiments on BERT Base (110M parameters) demonstrate the scalability and practical utility of the proposed Sassha method to larger models. The results seem to show notable improvements across multiple tasks. The performance gains are particularly meaningful, with accuracy improvements of 2.17% on RTE (71.48% vs SAM's 69.31%), substantial increases in MRPC accuracy/F1 scores (87.25%/90.97% vs SAM's 85.05%/89.82%), and consistent advantages in MNLI matched/mismatched accuracy (84.11%/85.05% vs SAM's 83.71%/83.82%). These improvements are particularly significant. I recommend the authors to incorporate all these additional experimental results in the revised manuscript, which would provide insights for the audience in the community.
> > > >
> > > > ---
> > > >
> > > > **(2) Regarding More Comparisons:**
> > > >
> > > > The direct comparisons with advanced SAM variants (ASAM and GSAM) appear to show competitive or superior performance across different network architectures and tasks. On CIFAR-10 with ResNet-32, Sassha achieves 94.093% accuracy, surpassing both ASAM (93.62%) and GSAM (93.76%). Similar advantages are observed on CIFAR-100, where Sassha reaches 72.143% compared to ASAM's 71.6% and GSAM's 72.10%. Even on the challenging ImageNet dataset with ViT architecture, Sassha maintains competitive performance (69.195%) with GSAM (69.600%). I recommend the authors to incorporate all these additional experimental results in the revised manuscript, which would provide insights for the audience in the community.
> > > >
> > > > ---
> > > >
> > > > **(3) Regarding the Technical Concerns:**
> > > >
> > > > The authors have also provided compelling clarifications for their technical choices, particularly the square root pre-conditioner. Their analysis spans multiple perspectives: the generalization benefits through optimal geometric interpolation, stability improvements near minima, and empirical validation through comprehensive ablation studies. The comparative analysis showing Sassha's superior performance (72.113%) over damping (71.27%) and clipping (69.47%) approaches provides strong empirical support for their design decisions. I recommend the authors to incorporate all these additional experimental results in the revised manuscript, which would provide insights for the audience in the community.
> > > >
> > > > ---
> > > >
> > > > **(4) Regarding the Practical Considerations:**
> > > > Regarding practical considerations, the authors have effectively addressed implementation concerns by demonstrating comparable tuning complexity to SAM+AdamW and providing empirical evidence for numerical stability. Their clarifications about memory requirements and practical implementation strategies make the method more accessible for real-world applications.
> > > >
> > > > ---
> > > >
> > > > However, one aspect warrants further attention: **the consistency of performance improvements across different tasks**. While the gains are broadly positive, their magnitude varies notably. For instance, on ImageNet with ResNet-50, the improvement is modest (Sassha: 76.429% vs GSAM: 76.450%), and some BERT fine-tuning tasks show mixed results. A deeper empirical analysis of these variations would provide valuable insights for practitioners in the community.
> > > >
> > > > Thus, my suggestions for revision are:
> > > >
> > > > - Incorporating all presented additional experimental results, especially the BERT Base experiments and comprehensive comparisons with SAM variants.
> > > > - Including the detailed theoretical analysis and ablation studies that clarify the advantages of the square root pre-conditioner.
> > > > - Adding the insightful discussions and clarifications in the responses to the revised manuscript to help the audience better understand this work.
> > > > - It will be valuable to conduct a thorough empirical analysis about performance variations across different tasks and architectures to further study the method’s generalizability.
> > > >
> > > > While some performance variations warrant further discussion, the overall contribution represents a meaningful advancement in second-order optimization. I look forward to further discussions with the authors.
> > > >
> > > > Best regards,
> > > >
> > > > Reviewer kwCD

---

> ### Author Response · Authors · 2024-12-04
> **Additional response**
>
> We deeply appreciate the reviewer for recognizing our efforts and leaving us a list of valuable suggestions and feedback, which will undoubtedly improve our work tremendously. We will ensure to incorporate all of these in the final version of the paper. Below, we address the reviewer’s last comment in detail.
>
> ---
>
> > A deeper empirical analysis of these variations would provide valuable insights for practitioners in the community.
>
> We sincerely thank the reviewer for the thoughtful suggestion, and we agree that this would be valuable to practitioners. Here, we present further empirical analysis and practical considerations for interpreting these variations in our results.
>
> **GSAM vs. Sassha.** While the accuracy difference between Sassha and GSAM may seem marginal in some cases (e.g., ImageNet), it is crucial to note that achieving comparable results with GSAM required us to perform significantly more hyperparameter tuning. In contrast, Sassha consistently achieved strong performance without these additional efforts.
>
> Specifically, GSAM introduces an additional hyperparameter $\alpha$ to penalize dominant eigenvalues, demanding as much tuning effort as tuning $\rho$. For instance, in the ResNet-50 setup, **tuning GSAM involved $8.75 \times$ larger search grids** compared to Sassha.
> `GSAM` (mostly follows the same search ranges presented in the GSAM paper)
> - ResNet on ImageNet:
>     - $\rho: \\{\textcolor{red}{0.01, 0.05}, 0.1, 0.15, 0.2, 0.25, \textcolor{red}{0.3}\\}$
>     - $\textcolor{red}{\alpha: \\{0.01, 0.05, 0.1, 0.15, 0.2\\}}$
>
> Additionally, we would like to provide our interpretation of how GSAM might achieve good performance in ImageNet. As discussed in [1], there could exist a discrepancy between the actual flatness (dominant Hessian eigenvalue) of the loss and the “perturbed loss” definition used by Sassha, which can worsen under a more complex and intricate loss landscape of ImageNet. Consequently, GSAM, designed to minimize dominant eigenvalues explicitly via a surrogate gap, may be better positioned to find flatter solutions than Sassha, which employs a sharpness minimization mechanism similar to SAM. Nevertheless, we note once again that **GSAM requires a much larger tuning budget than Sassha to achieve such results**.
>
>
> **BERT finetuning.** Here, while Sassha outperforms on subsets of tasks, it outperforms by larger margins compared to marginal differences in underperforming cases. Specifically, on tasks where Sassha outperforms, the performance gaps are more pronounced compared to that of underperforming cases ({+2.17%, +1.675%, +0.815% +0.23%} vs. {-0.11%, -0.035%, -0.11%}). Thus, we can say that on average, **Sassha outperforms in the BERT fintuning by +0.53%**. We additionally note that **it is common practice to evaluate GLUE finetuning tasks through their average score [2, 3]**.
>
> **Update on STSB.** The initial numbers we reported for the STSB results of Sassha were mistakenly put 84.86/84.76. This was a typo, and now we fixed it with correct results of 89.29/89.71.
>
> &nbsp;
>
> **Reference**\
> [1] Zhuang et al., Surrogate Gap Minimization Improves Sharpness-Aware Training. ICLR, 2022. \
> [2] Clark et al., ELECTRA: pre-training text encoders as discriminators rather than generator. ICLR, 2020. \
> [3] Hu et al., LORA: LOW-RANK ADAPTATION OF LARGE LANGUAGE MODELS. ICLR, 2022.

---

### Official Review · Reviewer_DoXq · 2024-11-04

**Soundness:** 3
**Presentation:** 3
**Contribution:** 2
**Rating:** 6
**Confidence:** 3

**Summary:**

This paper firstly investigates the solution sharpness of different second-order optimizers, pointing out that current second-order optimizers tend to converge to sharp solutions. Based on their findings, the authors combine SAM and second-order optimizers and design a sharpness-aware second-order optimizer, Sassha. Experiments show that Sassha can get better solutions than other second-order optimizers and is more robust to label noise than other second-order optimizers and SAM.

**Strengths:**

1. The perspective to investigate the sharpness of solutions obtained by second-order optimizers is novel, inspiring the community to design sharpness-aware second-order optimizers.
2. The findings of the lazy Hessian updates with SAM is interesting. I believe this deserves deeper investigations.
3. The robustness to label noise of Sassha is important.

**Weaknesses:**

1. The technical contribution of Sassha is minor, since Sassha just combine SAM and common second-order optimization techniques.


2. Although Sassha beats other second-order optimizers, its improvement seems incremental in standard training. The authors claims second-order optimizers converge faster than first-order optimizers, but Sassha is evaluated with the same training epochs (and Sassha needs one more forward-backward propagation). I thihk Table 6 shows the convergence advantage of Sassha, but the comparison only includes SAM and ViT model. Moreover, they did not compare SAM in stardard training, which seems that Sassha cannot performs better than SAM with same training budget.

    Overall, I think the authors should compare Sassha with other baselines including SAM under the same training time to validate the effectiveness.


3. It lacks deeper investigations why second-order optimziers converge to sharper solutions. The current paper just shows the phenomenon without any intuitive or theoretical results.

**Questions:**

1. In Table 7, why the average sharpness of SGD and SAM is negative values?

---

> ### Author Response · Authors · 2024-11-22
> **Response to DoXq**
>
> We appreciate the reviewer for taking the time to review our work. While we respond to the reviewer’s specific comments as below, please do let us know if there is anything else we can address further.
>
> ---
> **Sassha just combines SAM and common second-order optimization techniques?**
>
> We respectfully disagree with the reviewer. While the underlying idea behind Sassha may seem straightforward, it should not be judged without accounting for the technical and engineering challenges required to fully realize it. Extensive efforts were devoted to carefully engineering and examining various techniques, which all play a crucial role in mitigating specific issues arising from naively applying sharpness minimization to second order methods:
> - Absolute Hessian enables Sassha to avoid saddle points while preserving optimal rescaling of negative Hessians, unlike clipping or damping.
> - Square root is introduced to prevent divergent behavior from curvature under-estimation from introducing sharpness minimization.
> - Lazy Hessian updating is intentionally set to default to mitigate computation overhead based on investigations revealing that sharpness minimization can help enable this without severe performance degradation.
>
> We also note that this way of engineering is standard in optimization since ideas can achieve remarkable effectiveness when supported by seemingly simple but carefully considered techniques. For instance, Sophia has demonstrated that a simple clipping mechanism was sufficient to enhance second-order optimizers in training LLMs, gaining wide recognition by the community. Thus, reducing Sassha as merely combining SAM with common second-order optimizers neglects the complexities and efforts required to realize this concept, which is an unjust representation.
>
> To more directly support our claims, we compare Sassha to simple combinations of SAM + second-order optimizer (Sophia-H):
>
> ||RN20-CIFAR10|RN32-CIFAR10|RN32-CIFAR100|
> |-|-|-|-|
> |Sophia-H+SAM|92.53|93.59|71.312|
> |Sassha|**92.983**|**94.093**|**72.113**|
>
> We observe that it indeed underperforms Sassha given similar search space. This indicates that the techniques we designed for Sassha are non-trivial, and should be taken account of when evaluating Sassha.
>
> ---
> **Sassha vs. SAM & other first-order baselines under a fair setting**
>
> We appreciate the reviewer for the suggestion. We provide further comparison with SAM, please refer to Appendix G.2. For other first-order baselines, here we provide results for SGD and AdamW with twice the epoch budget than Sassha.
>
> ||RN20-CIFAR10|RN32-CIFAR10|RN32-CIFAR100|WRN28-CIFAR100|RN50-ImageNet|ViT_s-ImageNet|
> |-|-|-|-|-|-|-|
> |SGD/epoch|92.62/320|93.426/320|69.93/320|80.5/400|75.9/180|63.64/180|
> |AdamW/epoch|92.55/320|92.966/320|69.5/320|79.46/400|75.57/180|66.47/180|
> |Sassha/epoch|92.983/160|94.093/160|72.143/160|83.543/200|76.429/90|69.195/90|
>
> We observe that despite the first-order baselines being given more training budgets, they are still unable to outperform Sassha. We believe that this is due to the synergy between preconditioning and the bias towards flat solutions.
>
> ---
> **Lack of theoretical analysis on why SOO finds sharp solutions**
>
> We thank the reviewer for the insightful comment. Here we provide a theoretical account of why second-order methods might converge to sharp solutions as described in [2].
>
> According to the linear stability analysis of Wu et al. [2], SGD selects minima with the maximum Hessian eigenvalue ($\lambda_{\max}(H^\star)$, i.e., sharpness) satisfying the following condition:
> $$ \lambda_{\max}(H^\star)\leq2/\eta$$
> where the $\eta$ denotes the step size. This indicates that SGD escapes from sharp minima and converges to flat minima.
>
> However, approximate second-order optimizers (with preconditioner $D^{-1}\approx H^{-1}$) will converge to minima of the following condition:
> $$\lambda_{\max}(D^{-1}H^\star)\leq2/\eta$$
> From this, we observe that when $D^{-1}H^\star$ is close to the identity matrix, this condition is fulfilled as long as $\eta\leq2$. This implies that, unlike SGD, approximate second-order optimizers can converge to virtually any minima, including sharper ones that SGD might escape from.
>
> We will refer to these in the revised version.
>
> ---
> **Questions**
>
> > In Table 7, why the average sharpness of SGD and SAM is negative values?
>
> The formula we used for computing the average sharpness [3] is as follows:
> $$\mathbb{E}_{z\sim\mathcal{N}(0,1)}[L(x^{\star}+\rho z/\|z\|)-L{x^\star}],$$
> which can yield negative values if the solution $x^\star$ from the optimizer isn’t the exact minima of $L$, which can often be the case for deep learning.
>
> &nbsp;
>
> **Reference**\
> [1] Liu et al., Sophia: A Scalable Stochastic Second-order Optimizer for Language Model Pre-training, ICLR, 2024.\
> [2] Wu et al., How SGD Selects the Global Minima in Over-parameterized Learning: A Dynamical Stability Perspective. NeurIPS, 2018.\
> [3] Wen et al., How Sharpness-Aware Minimization Minimizes Sharpness? ICLR, 2023.

---

> > ### Comment · Reviewer_DoXq · 2024-11-25
> >
> > Thank you for your response. After revisiting the details of the method described in the paper, I now agree that the authors' approach to leveraging sharpness-aware minimization (SAM) in second-order optimization is meaningful. I apologize for not thoroughly comparing this aspect earlier. However, given that the improvements achieved by Sassha in the experiments are minor, I am raising my score to 6.
> >
> > Additionally, I believe the authors need to recheck the values in Table 7. Even if $x^\star$ is not the exact minimum within the neighborhood, it is unreasonable for the expected sharpness value to be negative — this implies that the loss value at $x^\star$ is higher than that of most solutions within the neighborhood. If a lower-loss solution can be easily found through random perturbations, it seems to indicate that the model has not been well-optimized.

---

> ### Author Response · Authors · 2024-11-28
> **Response to DoXq**
>
> We sincerely appreciate the reviewer for recognizing our contribution and raising the score. We are also grateful for the reviewer’s insightful feedback, which we believe will significantly enhance our paper. We will put our best efforts to reflect the reviewer's suggestions in the updated version.
>
> ---
>
> > it seems to indicate that the model has not been well-optimized
>
> Firstly, it is not quite likely that our models are not well-optimized, since their performance aligns well with previously reported results in prior works [1, 2].
>
> Even so, negative average sharpness values can naturally be measured for non-convex $L$ in deep learning (depending on the selected $\rho$). This is because the sharpness, by definition, simply measures the **loss difference** within a $\rho$-radius. This is slightly different from our earlier explanation (which was rooted from assuming a convex scenario), which we now see could be misleading. We apologize for the confusion and would like to clarify our stance. Please note that we have included this average sharpness measure alongside other metrics to provide a comprehensive reference as in prior work [3].
>
> Before we wrap up, in response to the reviewer’s comment on “minor improvements", we would like to gently highlight that the performance gains achieved by Sassha are in fact quite prominent. As shown in Table 2, Sassha consistently outperforms existing approximate second-order methods by at least 1%, and up to around 5%. Seeing Sassha exhibits improved performance across various tasks (even compared to SAM), we have a strong belief in the generality and robustness of Sassha.
>
> &nbsp;
>
> **Reference**\
> [1] Zagoruyko et al. Wide Residual Networks. CoRR, 2016. \
> [2] Kwon et al., ASAM: Adaptive Sharpness-Aware Minimization for Scale-Invariant Learning of Deep Neural Networks. ICML, 2021. \
> [3] Wen et al., How Sharpness-Aware Minimization Minimizes Sharpness? ICLR, 2023.

---

> > ### Author Response · Authors · 2024-12-04
> > **Closing remark**
> >
> > As the discussion period draws to a close, we express our sincere gratitude to the reviewer for their dedicated efforts and constructive comments. We promise to include detailed discussions, additional experiments, and suggestions in the final manuscript.

---

### Official Review · Reviewer_9BZk · 2024-11-04

**Soundness:** 3
**Presentation:** 4
**Contribution:** 2
**Rating:** 6
**Confidence:** 4

**Summary:**

The authors propose Sassha, a second-order optimizer combining ideas from sharpness-aware minimization and lazy second-order optimization. They provide experiments on image (CIFAR, ImageNet) and language (Glue, wikitext-2) tasks, reporting improved performance over baseline methods.

**Strengths:**

The paper is well-structured and written, and the overall presentation is good. It is easy to understand the central ideas and several questions I had when reading the paper were addressed shortly after with a suitable ablation study. The experimental results comparing Sassha to other second-order optimizers are convincing, and the gains over those baselines seem consistent. In general, the experimental evaluation is broad in terms of models and datasets. The ablation studies about why the lazy Hessian updates might benefit from SAM are insightful and support the claims made (although I have a few questions, see below).

**Weaknesses:**

**Novelty**

In my understanding, the presented method can be seen as applying SAM to Sophia-H as base-optimizer, with a few tweaks (using the square root and changing the clipping function of Sophia for bias correction). While I am unaware of work investigating SAM with second-order optimizers explicitly, it is well-known that adding SAM typically improves over the respective base-optimizers for many cases (e.g. SGD, Adam, AdamW, Lion, …) on the investigated datasets. I, therefore, think that it is not very surprising that also second-order methods benefit from SAM-like training. The main novelty and contribution of this work seem limited to the additional tweaks required when using second-order optimizers as SAM base optimizers. However, the performance of plain Sophia + SAM is not explored in comparison to Sassha.

**Comparison to SAM**

While Sassha improves over other second-order methods, the improvements over SAM-like methods are less clear. For Cifar, the results are relatively close, sometimes within standard deviations or even lower for the same computational budget (Table 14). For ImageNet, it would be good to see the models trained to convergence (e.g. 300 epochs). Additionally, I could not find the search space for the SAM parameter rho in these experiments. Assuming it matches the range reported for the label noise experiment (Table 13), I recommend trying slightly larger rho values and reporting results, ideally with accuracy vs. rho plots (as e.g. seen in Figure 5a of Becker et al. [1]), to allow a more thorough comparison between SAM and Sassha. For the language tasks, the comparisons to SAM are missing completely. Finally, to prove that Sassha is a sensible approach as a stand-alone method, comparisons to other SAM-like optimizers (e.g. [2], ...) would be necessary. In terms of presentation, I suggest including the SAM numbers in the Tables in the main paper alongside the second-order methods instead of the Appendix.

**Unclear motivation for the square root**

The choice to use the square root of the Hessian needs further explanation. The authors state that “underestimating curvature seems to be more prevalent under sharpness minimization,” but it is unclear if there are experiments directly supporting this claim. They suggest that training instabilities may result from certain entries approaching zero, but this alone does not justify using the square root, as similar stability could be achieved by adding a small scalar value or using clipping operations that preserve larger values, which might align better with the derivations in Section 4.1. A more detailed discussion of the square root’s impact on optimization—beyond mitigating instabilities from small Hessian entries—would be valuable. In the experiment on lazy Hessian training, both the square root and SAM are omitted, which (to my understanding) effectively reduces the algorithm to Sophia-H, with clipping instead of bias correction as the only difference.

[1] M. Becker, F. Altrock, and B. Risse, “Momentum-sam: Sharpness aware minimization without computational overhead,” arXiv preprint arXiv:2401.12033, 2024.

[2] J. Kwon, J. Kim, H. Park, and I. K. Choi, “Asam: Adaptive sharpness-
aware minimization for scale-invariant learning of deep neural net-
works,” in Proceedings of the International Conference on Machine
Learning (ICML).

**Questions:**

- Can the authors explain why Sophia-H and AdaHessian are so slow in terms of wall clock time (G.3)?
- Also, part of the speedup compared to Sophia-H might vanish when Sophia-H is used with a k-interval like Sassha. I think this should be stated more clearly.
- Is there a reason that Sophia-H is investigated, but Sophia-G is omitted?
- Did the authors use Shampoo or distributed Shampoo?


Minor remarks:

- In line 113 a reference is missing
- In Figure 5 the captions b) and c) don’t align with the y-axis label
- I suggest including the pseudo-code (Algorithm 1) in the main paper, since from Section 4 alone it is unclear what the final algorithm looks like.

---

> ### Author Response · Authors · 2024-11-22
> **Response to 9BZk (1/4)**
>
> We really appreciate the reviewer’s positive and constructive feedback. We believe this has led us to tremendously improve our work.  While we make clarifications to specific comments below, please let us know if there is anything we need to address further. We would be keen to engage in any further discussion.
>
> ---
>
> **Isn’t it just additional tweaks to second-order methods (or Sophia-H) and applying SAM?**
>
> We think it is reasonable to project that adding SAM can generally enhance the performance of the respective base optimizers. However, the process of implementing this effectively for second-order optimizers is far from trivial. Achieving seamless integration of sharpness minimization and second-order optimization to preserve their benefits while avoiding instability required careful engineering that involves exploration of various algorithmic design choices grounded in established prior studies (e.g., square root, absolute, see Section 4) and validating them through extensive testing (see Section 6).
>
> It is also worth noting that such engineering is a standard procedure in the literature since seemingly simple but well-chosen “tweaks” have been demonstrated to significantly improve performance, which are often appreciated by the community. Clipping, for instance, is a fundamental component in Sophia’s design and is, in fact, its sole feature distinguishing it from basic approximate second-order optimizers (just to be clear, bias correction has nothing to do with clipping, it is for correcting biases from zero-initializing first and second-moment variables).
>
> Nonetheless, we provide results (val accuracy) for simple Sophia-H + SAM on various tasks.
>
> ||RN20-CIFAR10|RN32-CIFAR10|RN32-CIFAR100|
> |-|-|-|-|
> |Sophia-H + SAM|92.530|93.590|71.312|
> |Sassha|**92.983**|**94.093**|**72.113**|
>
> These results indicate that Sassha indeed outperforms the combination of Sophia-H + SAM.  We attribute this to (i) the limitations of clipping and (ii) its reduced compatibility with SAM compared to the square root method.
>
> (i) The use of clipping results in Sophia partially performing signSGD over a subset of parameters [1], which may lead to suboptimal convergence in typical situations (on non-transformer architectures) [2, 3].
>
> (ii) When curvature becomes flat due to SAM, clipping may lead to a significant increase in the number of Hessian entries replaced by a very small constant $\epsilon$. This situation raises the sensitivity to hyperparameters like the clipping threshold and makes the optimization process more dependent on careful tuning.
>
> Conversely, square root does not enforce hard adjustments to the magnitude or direction of the update vector or to Hessian entries. Instead, it smoothly interpolate Hessian toward the identity matrix and performs Newton updates. A more detail explanation of the square root approach are provided in **on sqrt** below.
>
> For these reasons, we believe it would be unreasonable to consider Sassha as merely SAM with a second-order method as the base optimizer or Sophia with a few tweaks.
>
> ---
>
> **300 epochs for ImageNet**
>
> We appreciate the reviewer’s suggestion. The experiment is conducted as shown below,
>
> |Optimizer|Epoch|ViT_s_32-ImageNet|
> |-|-|-|
> |SAM (adamW) |300 | 68.808%|
> |Sassha|90 |69.195%|
> |Sassha|300 |69.418% |
>
> The result shows that Sassha remains consistently better, even with extended training epochs.
>
> ---
>
> **Large search space of rho**
>
> $\rho$ was searched over $\{0.01, 0.05, 0.1, 0.15, 0.2, 0.25, 0.3\}$, which is slightly larger than what was used in the label noise experiments. As the reviewer suggested, we extended this range to $0.8$ in increments of $0.05$. The results are as follows.
>
> *RN20 CIFAR10 (Sassha 92.983):*
>
> |rho|0.01|0.05|0.1|0.15|0.2|0.25|0.3|0.35|0.4|0.45|0.5|0.55|0.6|0.65|0.7|0.75|0.8|
> |-|-|-|-|-|-|-|-|-|-|-|-|-|-|-|-|-|-|
> |Val acc|92.377|92.743|92.703|92.847|92.823|92.547|92.257|92.023|91.79|91.123|90.807|90.36|90.04|89.563|89.307|88.953|88.463|
>
> *RN32 CIFAR10 (Sassha 94.093):*
>
> |rho|0.01|0.05|0.1|0.15|0.2|0.25|0.3|0.35|0.4|0.45|0.5|0.55|0.6|0.65|0.7|0.75|0.8|
> |-|-|-|-|-|-|-|-|-|-|-|-|-|-|-|-|-|-|
> |Val acc|92.377|92.743|92.847|93.723|93.893|93.597|93.57|93.253|93.027|92.807|92.573|92.137|91.9|91.153|90.717|90.29|89.333|89|
>
> *RN32 CIFAR100 (Sassha 72.143):*
>
> |rho|0.01|0.05|0.1|0.15|0.2|0.25|0.3|0.35|0.4|0.45|0.5|0.55|0.6|0.65|0.7|0.75|0.8|
> |-|-|-|-|-|-|-|-|-|-|-|-|-|-|-|-|-|-|
> |Val acc|69.91|70.383|71.143|71.9|71.993|71.89|71.747|71.763|71.357|71.143|70.66|70.03|69.64|68.85|68.537|67.89|66.617|
>
>
> We find that SAM still underperforms Sassha despite it being given a much larger search space (and hence increased search budget). We will include the search space and these results in the revised version.

---

> ### Author Response · Authors · 2024-11-22
> **Response to 9BZk (2/4)**
>
> **SAM for language tasks**
>
> We provide the validation performance of SAM on various language model tasks below.
>
> *Pretraining:*
>
> ||test ppl|
> |-|-|
> |SAM(AdamW)|158.06|
> |SASSHA|**122.40**|​
>
> *Finetuning:*
>
> |Task|RTE|STSB|MRPC|SST2|QQP|MNLI|QNLI|
> |-|-|-|-|-|-|-|-|
> |Metric|Acc|Spearman/Pearson|Acc/F1|Acc|Acc/F1|Mat / M.Mat|Acc|
> |SAM(AdamW)|72.56|88.98 / 89.31|84.56 / 88.89|90.83|90.124 / 86.785|**81.79 / 82.65**|89.07|
> |Sassha|**73.29**|**89.29 / 89.66**|**87.75 / 91.23**|**91.51**|**91.00 / 87.93**|81.74 / 82.12|**89.86**|
>
> We observe that Sassha outperforms SAM across most tasks. We will include this in the updated paper.
>
> ---
>
> **Comparisons to SAM variants**
>
> We conduct experiments to compare Sassha with advanced SAM variants [4-5]. We also evaluate G-Sassha (Sassha with surrogate gap guided sharpness from GSAM[5]) for fair comparison. The results are below.
>
> ||ASAM|GSAM|Sassha|G-Sassha|
> |-|-|-|-|-|
> |RN20 - CIFAR10|92.96%|92.71%|**92.983%**|92.943%|
> |RN32 - CIFAR 10|93.62%|93.76%|94.093%|**94.15%**|
> |RN32 - CIFAR 100|71.6%|72.10%|72.143%|**72.18%**|
> |RN50 - ImageNet|76.404%|**76.450%**|76.429%|*|
> |ViT_s - ImageNet|68.26%|69.600%|69.195%|**69.673%**|
>
> We find that Sassha is competitive with these advanced SAM variants. In addition, we note clearly that ASAM, according to its authors, may require a broader $\rho$ search range compared to the original SAM, leading to a more complex tuning process. Similarly, GSAM involves tuning additional hyperparameters that are critical yet quite sensitive.
>
> **Search space.** Here, we provide detailed setup and hyperparameter search space. For ResNet, we use SGD as the base optimizer for ASAM and GSAM, while for ViT, AdamW with gradient clipping set to 1.0 serves as the base optimizer. For all models, cross entropy loss is used, and the best learning rate and weight decay of the base optimizer are selected  in experiments with ASAM and GSAM. All algorithms are evaluated with fixed rho (not using rho scheduling). For ResNet on CIFAR, we apply multi-step decay with a decay rate of 0.1, and for ViT, we use cosine learning rate decay with 8 warm-up epochs.
> For GSAM and G-Sassha:
>
> - ResNet on CIFAR:
>     - $\rho$: {0.01, 0.05, 0.1, 0.15, 0.2, 0.25, 0.3, 0.35, 0.4}
>     - $\alpha$: {0.01, 0.05, 0.1, 0.15, 0.2, 0.25, 0.3}
> - ResNet on ImageNet:
>     - $\rho$: {0.01, 0.05, 0.1, 0.15, 0.2, 0.25, 0.3}
>     - $\alpha$: {0.01, 0.05, 0.1, 0.15, 0.2}
> - ViT on ImageNet:
>     - $\rho$: {0.1, 0.2, 0.3, 0.4, 0.5, 0.6}
>     - $\alpha$: {0.1, 0.2, 0.3}
>
> For ASAM:
> - $\rho$: {0.1, 0.15, 0.2, 0.3, 0.4, 0.5, 0.6, 0.7, 0.8, 0.9, 1, 1.5, 2}

---

> > ### Author Response · Authors · 2024-11-22
> > **Response to 9BZk (3/4)**
> >
> > **On square root**
> >
> > We greatly appreciate the reviewer’s constructive feedback.
> >
> > First, to examine whether the Hessian entries decrease (approaching zero) under sharpness minimization (SM), we compare the distribution of diagonal Hessian entries at different iterations (100, 150, and 200) for two configurations: **Baseline** ($=\text{Sassha} - \text{SM} - \text{Sqrt}$) and **Only-SM** ($=\text{Sassha} - \text{Sqrt}$, same as ‘No-sqrt’ in Figure 4), and see if Only-SM yields much smaller diagonal Hessian compared to the Baseline. However, since we cannot directly attach images in OpenReview, instead we present the range for the bottom 5% (min~5%ile) of the diagonal Hessian entries. We will provide the actual figure of distribution (e.g., Figure 4-(d)) in the revised paper.
> >
> > |iteration|100|150|200|
> > |-|-|-|-|
> > |Baseline|1.21e-09 ~ 1.31e-05|1.21e-09 ~ 1.30e-05|1.20e-09 ~ 1.30e-05|
> > |Only-SM|7.01e-10 ~ 8.88e-06|6.97e-10 ~ 8.90e-06|6.94e-10 ~ 9.64e-06|
> >
> > We observe that Only-SM does indeed possess smaller values in its diagonal Hessian entries compared to Baseline.
> >
> > Also, to evaluate whether damping (adding a small scalar value) and clipping ($max(h_t, \epsilon)$) can yield results similar to the square root (Sassha), we examine the distribution of the Hessian entries and their final validation accuracy.
> >
> > |iteration|100|150|200|
> > |-|-|-|-|
> > |Damping|1.083e-04 ~ 1.906e-04|1.109e-04 ~ 2e-04|1.133e-04 ~ 2.062e-04|
> > |Clipping |1e-04 ~ 1e-04| 1e-04 ~ 1e-04|1e-04 ~ 1e-04|
> > |Sassha|1.59e-05 ~ 9.99e-05|2.03e-05 ~ 1.20e-04|2.46e-05 ~ 1.38e-04|
> >
> > |Optimizer|RN32-CIFAR100|
> > |-|-|
> > |damping |71.27|
> > |clipping |69.47|
> > |Sassha |**72.113**|
> >
> > We see that while the two techniques manage to resolve numerical instability as predicted by the reviewer, their performance falls short of that achieved by Sassha.
> >
> > As to why this performance gap might be occurring, we make several hypotheses.
> >
> > First, we believe damping and clipping either shift the entire Hessian estimate or abruptly replace its certain entries to a predefined constant. Without careful tuning, these mechanisms may cause incorrect updates in specific directions, leading to performance degradation. In contrast, the square root adjusts Hessian entries in a gradual manner by interpolating them to the identity matrix, offering a smoother alternative to damping and clipping.
> >
> > Second, using square-rooted preconditioner can be seen as geometric interpolation between the first order method $H^{-\alpha}$, which has been demonstrated to allow selecting an optimal preconditioner that balances the bias and the variance of the population risk, thereby minimizing generalization error [6]. In general, $\alpha = 1/2$ (i.e., square root)  has shown moderate performance across various scenarios, and also provides additional benefits described below.
> >
> > Third, square root in particular can mitigate specific instabilities that approximate second-order optimizers suffer near minima. Specifically, since the commonly used diagonal Hessian estimators can scale inversely quadratic to the gradient, the preconditioned gradient update can scale inverse to the gradient which can cause instability as the gradient scale decreases as the model reaches near minima [7]. Square-rooting the preconditioner can safeguard such instabilities.
> >
> > These aspects potentially highlight additional positive effects of the square root approach beyond merely addressing training instability. We will include experimental results in the revised paper and discuss these points further.

---

> ### Author Response · Authors · 2024-11-22
> **Response to 9BZk (4/4)**
>
> **Questions**
>
> > Can the authors explain why Sophia-H and AdaHessian are so slow in terms of wall clock time (G.3). Also, part of the speedup compared to Sophia-H might vanish when Sophia-H is used with a k-interval like Sassha. I think this should be stated more clearly.
>
> We thank the reviewer for the observation regarding the wall clock time differences.
>
> The longer wall clock time of Sophia-H and AdaHessian compared to Sassha arises from our decision not to apply the lazy Hessian update to both algorithms in the experiments. This choice was made to compare Sassha against the two algorithms under their best performance configurations. Specifically, Liu et al. [1] demonstrated that Sophia achieves its peak performance at k=1, and AdaHessian [8] does not support lazy Hessian updates in its original design.
>
> We believe this to be fairer as our claim for Sassha focuses on enhancing the generalization performance of second-order optimizers, not their efficiency. Introducing lazy Hessian updates for Sophia in our main experiments would improve speed but at the cost of reduced performance compared to what is currently reported in our paper, which still underperforms ours. Nonetheless, we will make this point clearer in the revision.
>
> > Is there a reason that Sophia-H is investigated, but Sophia-G is omitted?
>
> We would first like to clarify that ‘H’ and ‘G’ refer to Hutchinson’s unbiased diagonal Hessian estimator and the Gauss-Newton-Bartlett biased diagonal Hessian estimator, respectively, for computing the preconditioner. However, as noted by Liu et al. [1], there is no discernable performance difference between the two.
>
> In our experiments, we used Sophia-H to ensure a consistent comparison of the effects of sharpness minimization while keeping the diagonal Hessian estimator identical.
>
> > Did the authors use Shampoo or distributed Shampoo?
>
> We used the original Shampoo [9] for our experiments.
>
> ---
>
> **Minor remarks**
>
> We appreciate the reviewer’s detailed feedback and suggestions. We will make the necessary adjustments in the revised paper.
>
> &nbsp;
>
> **Reference**\
> [1] Liu et al., Sophia: A Scalable Stochastic Second-order Optimizer for Language Model Pre-training, ICLR, 2024. \
> [2] Kunstner et al., Noise Is Not the Main Factor Behind the Gap Between SGD and Adam on Transformers, but Sign Descent Might Be. ICLR, 2023. \
> [3] Karimireddy et al., Error Feedback Fixes SignSGD and other Gradient Compression Schemes. ICML, 2019. \
> [4] Kwon et al., ASAM: Adaptive Sharpness-Aware Minimization for Scale-Invariant Learning of Deep Neural Networks. ICML, 2021. \
> [5] Zhuang et al., Surrogate Gap Minimization Improves Sharpness-Aware Training. ICLR, 2022. \
> [6] Amari et al., When does Preconditioning Help or Hurt Generalization? ICLR, 2021. \
> [7] Kunstner et al., Limitations of the Empirical Fisher Approximation for Natural Gradient Descent. NeurIPS, 2019. \
> [8] Yao et al., ADAHESSIAN: An Adaptive Second Order Optimizer for Machine Learning.  AAAI, 2020. \
> [9] Gupta et al., Shampoo: Preconditioned stochastic tensor optimization. ICLR, 2018.

---

> > ### Comment · Reviewer_9BZk · 2024-11-26
> > **Keeping my score**
> >
> > I appreciate the authors' effort in their extensive rebuttal. Below I address specific points:
> >
> > **Novelty**: I agree with the authors that well-chosen “tweaks” and careful design choices can be a valuable contribution and that engineering can be far from trivial, but it also has to be shown that it is indeed far from trivial. The experiment on Sophia-H + SAM indicates that even a naive combination of existing second-order optimizers with SAM might already achieve fairly good performance, at least compared to other second-order optimizers. Therefore, the primary goal of SASSHA (“ enhancing the generalization of approximate second-order methods efficiently”) could have been achieved without “tweaks” or “careful design choices”. This is not to say that SASSHA is not potentially useful (it outperforms Sophia-H + SAM in the three CIFAR tasks), but that the framing of the central messages of the paper needs to be careful. In essence, it underlines that for SASSHA to be a valid contribution, it needs to improve over SAM-baselines, as improvements over second-order optimizers could have been achieved by a naive baseline.
> >
> >
> > **Comparison to SAM**:
> > I appreciate the additional experiments with ASAM and GSAM on CIFAR/ImageNet, and with SAM on the language tasks, as well as the increased search space for $\rho$. While I can see improvements of SASSHA for certain scenarios (e.g. language pretraining, ViT on ImageNet), differences are often small (CIFAR, esp. with the newer SAM variants) or inconsistent (Bert-Base ft, RN50 ImageNet).
> >
> >
> > **Unclear motivation for square root:**
> > I appreciate the additional experiments, especially on clipping and damping. An alternative interpretation of those results could be that the square root has beneficial properties that align well with SAM-training and go beyond mitigating small diagonal entries in the Hessian (I don't think further experiments on this are necessary, but would like to encourage the authors to be careful in their claims in the paper).
> >
> >
> > One more remark: For ViT-S32, I’m surprised by the low numbers of SAM+AdamW. For almost identical settings (300 epochs, cosine lr decay, lr 0.001, 1024 batch size, 8 epochs warmup), numbers that are >2.5% larger are achieved (e.g., in [1]). I’m unsure about the reason for this, but I encourage the authors to double-check their implementation carefully.
> >
> >
> > **Summary**: I appreciate the authors' effort and the additional insights the rebuttal brought. However, my doubts about SASSHA as a stand-alone method that improves over other SAM-like optimizers remain. I, therefore, will keep my score.
> >
> >
> > [1] M. Becker, F. Altrock, and B. Risse, “Momentum-sam: Sharpness aware minimization without computational overhead,” arXiv preprint arXiv:2401.12033, 2024

---

> ### Author Response · Authors · 2024-12-03
> **Additional response (1/2)**
>
> We really appreciate the reviewer’s continuing engagement in our work. We address the specific points below.
>
> ---
>
> **Novelty**
>
> > In essence, it underlines that for SASSHA to be a valid contribution, it needs to improve over SAM-baselines, as improvements over second-order optimizers could have been achieved by a naive baseline.
>
> First, we would like to note that the performance improvements of Sassha over naive baselines (Sophia-H + SAM) should not be overlooked, as they are noticeable and consistent. For instance, Sassha outperforms Sophia-H + SAM by +0.453%p on ResNet20/CIFAR-10, which is more noticeable compared to widely recognized ASAM which improves over its baseline (SAM) by +0.26%p on the same task [2]. Also note that Sassha shows mostly competitive performance or even surpasses SAM in various cases, whereas Sophia-H + SAM underperforms SAM.
>
> More importantly, **achieving these results with Sophia-H + SAM required extensive hyperparameter search to tune its additional clipping threshold and $\epsilon$ ($28\times$ larger search grids in our setup)**, as the value suggested in the original Sophia paper [3] does not generalize well beyond language tasks. In contrast, Sassha consistently achieves robust performance across all tested tasks without any additional tuning, as it shares the same set of hyperparameters as SAM.
>
> Thus, with respect, we find the critique (“the improvements can be achieved by a naive baseline”) too stretched to accept, since Sassha (1) consistently outperforms Sophia-H + SAM, and (2) it does so without requiring extra extensive hyperparameter tuning.
>
>
> ---
>
> **Comparison to SAM**
>
> > While I can see improvements of SASSHA for certain scenarios (e.g. language pretraining, ViT on ImageNet), differences are often small (CIFAR, esp. with the newer SAM variants) or inconsistent (Bert-Base ft, RN50 ImageNet).
>
> We relate to the reviewer’s concern. However, we find it a bit unfairly harsh, by being only focused on “performance” without concerning budgets (for newer SAM variants) or partly incorrect (for Bert-Base ft).
>
> First, SAM variants require considerably more hyperparameter tuning to achieve generalization performance comparable to Sassha. For example, GSAM introduces an additional hyperparameter $\alpha$, demanding as much tuning effort as tuning $\rho$. Similarly, ASAM, as noted by its authors, typically necessitates exploring a broader $\rho$ range, as its appropriate value is approximately 10 times larger than that of SAM. In our setup, **tuning GSAM and ASAM involved  $4.5\times\sim15.75\times$ and $3\times$ larger search grids** compared to Sassha, respectively.
>
> `GSAM` (mostly followed the same search ranges presented in the GSAM paper)
> - ResNet on CIFAR:
>     - $\rho: \\{\textcolor{red}{0.01, 0.05}, 0.1, 0.15, 0.2, 0.25, \textcolor{red}{0.3, 0.35, 0.4}\\}$
>     - $\textcolor{red}{\alpha: \\{0.01, 0.05, 0.1, 0.15, 0.2, 0.25, 0.3\\}}$
> - ResNet on ImageNet:
>     - $\rho: \\{\textcolor{red}{0.01, 0.05}, 0.1, 0.15, 0.2, 0.25, \textcolor{red}{0.3}\\}$
>     - $\textcolor{red}{\alpha: \\{0.01, 0.05, 0.1, 0.15, 0.2\\}}$
> - ViT on ImageNet:
>     - $\rho: \\{0.1, 0.2, 0.3, 0.4, \textcolor{red}{0.5, 0.6}\\}$
>     - $\textcolor{red}{\alpha: \\{0.1, 0.2, 0.3\\}}$
>
> `ASAM`
> - $\rho: \\{0.1, 0.15, 0.2, 0.3, \textcolor{red}{0.4, 0.5, 0.6, 0.7, 0.8, 0.9, 1.0, 1.5}\\}$
>
> Next, it appears that the reviewer may have misinterpreted the fine-tuning results (on squeezeBERT) in our first response since Sassha consistently outperforms SAM across almost all tasks. Notably, Sassha achieves average improvements of approximately 3% in MRPC and 1% in QQP.

---

> ### Author Response · Authors · 2024-12-03
> **Additional response (2/2)**
>
> **ViT results**
>
> > For ViT-S32, I’m surprised by the low numbers of SAM+AdamW. For almost identical settings (300 epochs, cosine lr decay, lr 0.001, 1024 batch size, 8 epochs warmup), numbers that are >2.5% larger are achieved (e.g., in [1]). I’m unsure about the reason for this, but I encourage the authors to double-check their implementation carefully.
>
> Upon investigation, we found key differences in the setup that may have resulted in this performance difference. Specifically, the official implementation for MSAM [4] revealed that they set the ViT depth to 16 and employed advanced methods such as cutout regularization and rho scheduling. By contrast, we configured the ViT depth with 12 following the original ViT paper and its variants [5-7] and did not use the aforementioned additional techniques. We also note that our results are the average of three random seeds, whereas their results are based on a single run. Nevertheless, **we believe this does not affect the conclusion of our results, since all algorithms were compared fairly in the identical experiment settings**.
>
> ---
>
> **Final remark**
>
> We sincerely appreciate the time and effort the reviewer dedicated to providing such thoughtful feedback. With respect, however, we find that some comments are quite focused on “performance perspective”, thereby potentially overlooking other significant contributions presented in our work, most notably for example:
> - We conducted extensive experiments to investigate the correlation between the poor generalization of approximate second-order methods and the sharpness of their solution. To the best of our knowledge, no prior work has explicitly investigated this to any degree matching ours.
> - We developed a new practical and robust second-order method that produces strong and consistent results across diverse sets of deep learning experiments, which is validated through a series of ablation analyses, and also, theoretically proven with a convergence analysis.
> - We analyzed the positive impact of sharpness minimization on Hessian reuse. Our empirical investigations suggest that this effect may stem from sharpness minimization slowing the rate of change in the Hessian between iterations. This observation highlights sharpness minimization as a promising approach to addressing the inefficiencies associated with Hessian computation in second-order methods.
>
> We hope these contributions are appropriately recognized and considered in the evaluation of our work.
>
> &nbsp;
>
> **Reference**\
> [1] Becker et al., “Momentum-sam: Sharpness aware minimization without computational overhead,” arXiv, 2024. \
> [2] Kwon et al., ASAM: Adaptive Sharpness-Aware Minimization for Scale-Invariant Learning of Deep Neural Networks. ICML, 2021. \
> [3] Liu et al., Sophia: A Scalable Stochastic Second-order Optimizer for Language Model Pre-training, ICLR, 2024. \
> [4] MarlonBecker, [Official implementation of MSAM](https://github.com/MarlonBecker/MSAM/blob/main/configs/ImageNet_ViT.ini), 2024. \
> [5] Dosovitskiy et al., An image is worth 16x16 words: Transformers for image recognition at scale. ICLR, 2021. \
> [6] Xu et al., Supplementary to GroupViT: Semantic Segmentation Emerges from Text Supervision. CVPR, 2022. \
> [7] Touvron et al., Training data-efficient image transformers & distillation through attention. ICML, 2021.

---

> > ### Author Response · Authors · 2024-12-04
> > **Closing remark**
> >
> > As the discussion period draws to a close, we would like to sincerely thank the reviewer for dedicating their time and effort to reviewing our paper and providing constructive feedback. We will make sure to include all suggestions and additional experiments in the final version.

---

### Comment · Area_Chair_aJ5f · 2024-11-23
**Request for Further Discussions**

Dear PC Members,

Thank you for your valuable comments during the review period, which raised many interesting and insightful questions. The authors have now posted their feedback, and I encourage you to review their responses and engage in further discussion if necessary.

I understand that you may have a busy schedule, but your additional input is highly appreciated, particularly for papers that are initially on the borderline. Your contributions are crucial in ensuring a fair and well-rounded decision-making process.

Thank you once again for your continued support and dedication to ICLR.

Best regards,
AC

---

### Comment · Area_Chair_aJ5f · 2024-12-03
**A quick question**

Dear Authors,

I am glad to see  active discussions during the rebuttal stage. I have a quick question regarding a closely related work:

Li et al., Enhancing Sharpness-Aware Optimization Through Variance Suppression.

While the starting points of their work and yours may differ, the approaches seem quite similar, as both use exponential moving averages rather than gradients to achieve better adversarial directions.

Could you clarify the distinctions between your approach and theirs in more detail? Additionally, I noticed a discrepancy in the reported performance. For instance, with ViT-S/32 on ImageNet, the vanilla training in Li et al. achieves 68.12%, whereas your paper reports only 62.94%.

Thank you, and I look forward to your response.

Best Regards,
AC

---

> ### Author Response · Authors · 2024-12-04
>
> We sincerely thank the AC for their time and efforts to actively participate in reviewing our paper, encouraging discussions, and sharing feedback that would enrich the quality of our work. We address your question below.
>
> ---
>
> **Compare to Li et al. [1]**
>
> With respect, we believe that there are major differences between our work and [1], which we detail below.
>
> First, the purpose of [1] differs significantly from Sassha. Precisely, Sassha aims to mitigate the poor generalization issue of second-order optimizers by incorporating sharpness minimization. In contrast, [1] focuses on improving the adversarial step in SAM by addressing the issue of an “overly-friendly adversary” via variance suppression.
>
> Consequently, their resulting algorithms share little similarity. Specifically, Sassha, designed around approximate second-order optimization, employs **diagonal Hessian preconditioning** in its descent step. Furthermore, as its name suggests, Sassha introduces **techniques (e.g., square root and absolute function) to ensure stable Hessian approximation**. Meanwhile, Vasso [1], built around SAM, introduces an exponential moving average (EMA) to its adversarial step for efficient variance suppression.
>
> Thus, the adversarial direction used by Sassha differs from that of [1], since Sassha utilizes the non-EMA perturbation step as in the original SAM.
>
> We summarize these differences in the following table:
>
> ||EMA (for perturbation)|pertubation|update rule|
> |-|-|-|-|
> |**Vasso [1]**|$\textcolor{red}{d_{t+1} = (1-\theta)d_t + \theta \nabla f_t(x_t)}$|$\epsilon = \rho\frac{\textcolor{red}{d_t}}{\lVert \textcolor{red}{d_t}\rVert} $| $x_{t+1} = x_t - \eta\nabla f_t(x_t+\epsilon)$|
> |**Sassha**|-|$\epsilon = \rho\frac{\nabla f_t(x_t)}{\lVert \nabla f_t(x_t)\rVert}$ | $ x_{t+1} = x_t - \eta \textcolor{blue}{\sqrt{\lvert H_t(x_t+\epsilon) \rvert}^{-1}} \nabla f_t(x_t+\epsilon)$|
>
> ---
>
> **Discrepancy in ViT-S/32 performance**
>
> We’re really grateful for your observation on this matter.
>
> Upon closer inspection, we find this discrepancy to arise primarily from differences in experimental configurations. Specifically, $\textcolor{red}{\text{[1]}}$ reports results using the more advanced $\textcolor{red}{\textbf{AdamW}}$ optimizer with a total of $\textcolor{red}{\textbf{300 epochs}}$ for optimization. In contrast, $\textcolor{blue}{\text{our vanilla training}}$ result of 62.94% was achieved using $\textcolor{blue}{\textbf{SGD}}$ with a significantly shorter training duration of $\textcolor{blue}{\textbf{90 epochs}}$.
>
> For reference, we also provide results using AdamW. With 90 epochs, our method achieves 66.46% accuracy, and we anticipate that with 300 epochs—similar to [1]—our results would align closely with theirs.
>
> However, more importantly, **we believe this would not undermine the validity of our conclusion, since all comparisons between Sassha and other methods are conducted under identical experiment settings**.
>
> ---
>
> **Final remark**
>
> Once again, we really appreciate the AC for sparing their valuable time to review our work.  We hope that our findings and discussions prove insightful and that our contributions can be acknowledged as recognized by most of the reviewers.
>
> &nbsp;
>
> **Reference**\
> [1] Li et al.,  Enhancing Sharpness-Aware Optimization Through Variance Suppression, Neurips, 2023.

---

### Meta-Review · Area_Chair_aJ5f · 2024-12-19

**Metareview:**

This paper proposes using second-order information to reduce the "overly-friendly adversary." All reviewers agree that the method introduces a novel perspective, particularly appreciating the findings related to the lazy Hessian. However, the paper has notable weaknesses. These include a less significant technical contribution, as it simply combines SAM with second-order information estimation. More importantly, the numerical experiments raised many concerns, such as relatively low accuracy for other methods and insufficient training of baselines (i.e., missing validation with 300-epoch training). The latter is especially important since this paper focuses on improving generalization capability—starting from an inadequately trained solution cannot effectively demonstrate generalization benefits.

After active and insightful discussions, the paper ultimately received three scores of 6 and one score of 5. However, two reviewers clarified that their scores of 6 are borderline, with 5-6 being a more accurate reflection.

Thus, the overall score is approximately 5.5, which aligns with my evaluation. To ensure a fair evaluation, a thorough discussion with the SAC was conducted, and the reviewers' opinions were discussed during this process. Based on all the feedback, I recommend rejecting the paper. I hope this decision will not discourage the authors from improving their work, and I trust the discussions have been constructive and helpful for further development.

**Additional Comments On Reviewer Discussion:**

This paper garnered significant attention from the reviewers, leading to active and constructive discussions among the authors, reviewers, and myself.

The discussions primarily focused on the experiments. Frankly, the original experiments were quite weak, with reviewers raising numerous valid concerns. The authors made considerable efforts to provide additional experiments during the rebuttal stage, which partially addressed the reviewers' concerns. As a result, several reviewers increased their scores.

However, some concerns remain unresolved. Notably, two reviewers indicated that their scores of 6 were borderline, leaning more towards 5-6. Additionally, Reviewer kwCD remarked, "many crucial experimental results were only provided during the rebuttal stage rather than in the original manuscript," highlighting the authors' substantial effort during the rebuttal process but also underscoring that the original submission was not well-prepared.

Overall, there is a consensus that this paper falls on the borderline. To ensure a fair evaluation, a thorough discussion with the SAC was conducted, and the reviewers' opinions were confirmed during this process. Based on the collective feedback, I recommend rejecting the paper.

---

### Decision · Program_Chairs · 2025-01-22

Reject